# AutoGEL: An Automated Graph Neural Network with Explicit Link Information

**Zhili WANG, Shimin DI*, Lei CHEN**
Department of Computer Science and Engineering
The Hong Kong University of Science and Technology
`{zwangeo,sdiaa,leichen}@connect.ust.hk`

## Abstract

Recently, Graph Neural Networks (GNNs) have gained popularity in a variety of real-world scenarios. Despite the great success, the architecture design of GNNs heavily relies on manual labor. Thus, automated graph neural network (AutoGNN) has attracted interest and attention from the research community, which makes significant performance improvements in recent years. However, existing AutoGNN works mainly adopt an implicit way to model and leverage the link information in the graphs, which is not well regularized to the link prediction task on graphs, and limits the performance of AutoGNN for other graph tasks. In this paper, we present a novel AutoGNN work that explicitly models the link information, abbreviated to AutoGEL. In such a way, AutoGEL can handle the link prediction task and improve the performance of AutoGNNs on the node classification and graph classification task. Specifically, AutoGEL proposes a novel search space containing various design dimensions at both intra-layer and inter-layer designs and adopts a more robust differentiable search algorithm to further improve efficiency and effectiveness. Experimental results on benchmark data sets demonstrate the superiority of AutoGEL on several tasks.

## 1 Introduction

As one of ubiquitous data structures, graph $G(E, V)$ contains the node-set $V = \{v_1, \cdots, v_n\}$ and edge-set $E = \{e(v_i, v_j) : v_i, v_j \in V\}$, which can represent a lot of real-world data sets, such as social networks [Bu et al., 2018], physical systems [Sanchez-Gonzalez et al., 2018], protein-protein interaction networks [Yue et al., 2020]. In recent years, Graph Neural Networks (GNNs) have been introduced for various graph tasks and achieve unprecedented success, such as node classification [Kipf and Welling, 2016a, Hamilton et al., 2017], link prediction [Vashishth et al., 2019, Li et al., 2020], and graph classification [Niepert et al., 2016, Zhang et al., 2018]. Generally, GNNs encode $G(V, E)$ into the $d$-dimensional vector space (e.g., $\mathbf{V} \in \mathbb{R}^{|V| \times d}$) that preserves similarity in the original graph. Despite the great success, those GNNs are restricted to specific instances within GNN design space [You et al., 2020]. Different graph tasks usually require different GNN architectures [Gilmer et al., 2017]. For example, compared with the node classification task, GNNs for graph classification introduces an extra readout phase to obtain graph embeddings. However, architecture design for these GNNs remains a challenging problem due to the diversity of graph data sets. Given a graph task, a GNN architecture performs well on one data set does not necessarily imply that it is also suitable for other data sets [You et al., 2020].

Some pioneer works have been proposed to alleviate the above issue in GNN models. They introduce Neural Architecture Search (NAS) [Elsken et al., 2019] approaches to automatically design suitable

---

*Corresponding author

35th Conference on Neural Information Processing Systems (NeurIPS 2021).

GNN architecture for the given data set (i.e., AutoGNN) [Zhou et al., 2019, Gao et al., 2020, Jiang and Balaprakash, 2020, Zhao et al., 2021]. Architectures identified by these AutoGNNs rival or surpass the best handcrafted GNN models, demonstrating the potential of AutoGNN towards better GNN architecture design. Unfortunately, the existing AutoGNNs are mainly designed for the node classification and graph classification task. Their designs do not include edge embeddings, i.e., modeling and organizing link information in an implicit way. First, it is difficult for existing AutoGNNs to handle another important graph task on the edge-level, link prediction (LP) task. Second, lack of edge embedding makes them inexpressive to leverage the complex link information, such as direction information of edges and different edge types in multi-relational graphs. Especially, various edge types could impose different influence for encoding nodes into embeddings, which can further improve the model performance on the node-level and graph-level tasks. Therefore, a new AutoGNN is desired to model link information explicitly on various data sets.

To bridge the aforementioned research gap, we propose AutoGEL, a novel *AutoG*NN framework with *E*xplicit *L*ink information, which can handle the LP task and improve performance of AutoGNNs on other graph tasks. Specifically, AutoGEL explicitly learns the edge embedding in the message passing framework to model the complex link information, and introduces the several novel design dimensions into the GNN search space, enabling a more powerful GNN to be searched for any given graph data set. Moreover, AutoGEL adopts a robust differentiable search algorithm to guarantee the effectiveness of searched architectures and control the computational footprint. We summarize the contributions of this work as follows:

- The design of existing AutoGNNs follows an implicit way to leverage and organize the link information, which cannot handle the LP task and limits the performance of AutoGNNs on other graph tasks. In this paper, we present a novel method called AutoGEL to solve these issues through explicitly modeling the link information in the graphs.

- AutoGEL introduces several novel design dimensions into the GNN search space at both the intra-layer and inter-layer designs, so as to improve the task performance. Moreover, motivated by one robust NAS algorithm SNAS, AutoGEL upgrades the search algorithm adopted in existing AutoGNNs to further guarantee the effectiveness of final derived GNN.

- The experimental results demonstrate that AutoGEL can achieve better performance than manually designed models in the LP task. Furthermore, AutoGEL shows excellent competitiveness with other AutoGNN works on the node and graph classification tasks.

## 2 Related Work

### 2.1 General Message Passing Framework

The majority of GNNs follow the neighborhood aggregation schema [Gilmer et al., 2017], i.e., the Message Passing Neural Network (MPNN), which is formulated as:

$$\mathbf{m}_v^{k+1} = AGG_k(\{M_k(\mathbf{h}_v^k, \mathbf{h}_u^k, \mathbf{e}_{vu}^k) : u \in N(v)\}), \ \mathbf{h}_v^{k+1} = ACT_k(COM_k(\{\mathbf{h}_v^k, \mathbf{m}_v^{k+1}\})), \quad (1)$$

$$\hat{\mathbf{y}} = R(\{\mathbf{h}_v^L | v \in G\}), \quad (2)$$

where $k$ denotes $k$-th layer, $N(v)$ denotes a set of neighboring nodes of $v$, $\mathbf{h}_v^k$, $\mathbf{h}_u^k$ denotes hidden embeddings for $v$ and $u$ respectively, $\mathbf{e}_{vu}^k$ denotes features for edge $e(v, u)$ (optional), $\mathbf{m}_v^{k+1}$ denotes the intermediate embeddings gathered from neighborhood $N(v)$, $M_k$ denotes the message function, $AGG_k$ denotes the neighborhood aggregation function, $COM_k$ denotes the combination function between intermediate embeddings and embeddings of node $v$ itself from the last layer, $ACT_k$ denotes activation function. Such message passing phase in (1) repeats for $L$ times (i.e., $k \in \{1, \cdots, L\}$). For graph-level tasks, it further follows the readout phase in (2) where information from the entire graph $G$ is aggregated through readout function $R(\cdot)$.

### 2.2 Automated Graph Neural Networks (AutoGNN)

In recent years, AutoGNN has emerged as a promising direction towards better graph neural architecture design [Zhou et al., 2019, Gao et al., 2020, Jiang and Balaprakash, 2020, You et al., 2020, Zhao et al., 2021, Ding et al., 2021]. To enable a powerful GNN architecture to be searched, AutoGNNs first propose the GNN search space in the *intra-layer* level, i.e., providing common choices for several

Table 1: Overview of Existing AutoGNN Works. The "Differ." denotes to differentiable algorithm.

| Method | Graph | MPNN Space | | | Search Algorithm | Task |
|---|---|---|---|---|---|---|
| | #node/edge types | $\mathbf{h}_e$ | intra | inter | | |
| GraphNAS | $\geq 1/=1$ | $\times$ | $\checkmark$ | $\times$ | RL | node |
| AGNN | $\geq 1/=1$ | $\times$ | $\checkmark$ | $\times$ | EA+RL | node |
| SANE | $\geq 1/\geq 1$ | $\times$ | $\checkmark$ | $layer\_cnt\&\_agg$ | deterministic Differ. | node |
| NAS-GCN | $\geq 1/\geq 1$ | $\times$ | $\checkmark$ | $layer\_cnt$ | EA | graph |
| You et al. [2020] | $\geq 1/=1$ | $\times$ | $\checkmark$ | $layer\_cnt$ | random | node/edge/graph |
| AutoGEL | $\geq 1/\geq 1$ | $\checkmark$ | $\checkmark$ | $layer\_cnt\&\_agg$ | stochastic Differ. | node/edge/graph |

important operators in one MPNN layer (see (1) and (2)). Here we summarize candidate choices for those operators:

- **Message Function** $M_k$: Existing AutoGNNs mainly focus on the node-level and graph-level task, thus edge embeddings are often not available. $M_k(\mathbf{h}_v, \mathbf{h}_u, \mathbf{e}_{vu})$ is reduced to $M_k(\mathbf{h}_v, \mathbf{h}_u)$. Typically, $M(\mathbf{h}_v, \mathbf{h}_u) = a_{vu}\mathbf{W}\mathbf{h}_u$, where $a_{vu}$ denotes the attention scores, and $\mathbf{W} \in \mathbb{R}^{d \times d}$ denotes the linear transformation matrix. Note that NAS-GCN [Jiang and Balaprakash, 2020] only takes the edge feature $\mathbf{e}_{vu}$ as input without learning edge embeddings. We next denote the edge embedding to $\mathbf{h}_e$ for distinguishing.

- **Aggregation** $AGG_k$: It controls the way to aggregate message from nodes' neighborhood. It can be any differentiable and permutation invariant functions, usually $AGG_k \in [sum, mean, max]$. And $sum(\cdot) = \sum_{u \in N(v)} M_k(\mathbf{h}_v, \mathbf{h}_u)$, $mean(\cdot) = \sum_{u \in N(v)} M_k(\mathbf{h}_v, \mathbf{h}_u)/|N(v)|$, and $max(\cdot)$ denotes channel-wise maximum across the node dimension.

- **Combination** $COM_k$: It determines the way to merge messages between neighborhood and node itself. In literature, $COM_k$ is selected from $[concat, add, mlp]$, where $concat(\cdot) = [\mathbf{h}_v^k, \mathbf{m}_v^{k+1}]$, $add(\cdot) = \mathbf{h}_v^k + \mathbf{m}_v^{k+1}$, and $mlp(\cdot) = \mathbf{MLP}(\mathbf{h}_v^k + \mathbf{m}_v^{k+1})$ (**MLP** is Multi-layer Perceptron).

- **Activation** $ACT_k$: $[identity, sigmoid, tanh, relu, elu]$ are some of the most commonly used activation functions [Gao et al., 2020].

- **Graph Pooling:** The pooling operator has been introduced for the graph classification task, such as $[global\ pool, global\ attention\ pool, flatten]$ [Jiang and Balaprakash, 2020, Wei et al., 2021].

In addition to the above intra-layer operators, several works [You et al., 2020, Zhao et al., 2021, Jiang and Balaprakash, 2020] propose the idea of searching layer connectivity to combine hidden representations of different layers in a better way, i.e., *inter-layer* design. More details will be discussed in Sec. 3.1.2. After the search space design, AutoGNNs adopt various search algorithms to search the optimal architecture from the search space. AGNN [Zhou et al., 2019] and GraphNAS [Gao et al., 2020] follow the reinforcement learning (RL) [Williams, 1992] way to search architectures. They utilize a recurrent neural network controller for architecture sampling, and update the controller to maximize the expected performance of sampled architectures. NAS-GCN [Jiang and Balaprakash, 2020] adopt evolutionary algorithm (EA), where new architectures are generated by performing mutation from parent architectures and the population, i.e., the best performing architectures, are iteratively updated. However, both RL and EA algorithms require a large number of architectures to be sampled, which is inherently computational expensive. To improve the search efficiency, SANE [Zhao et al., 2021] adopts a deterministic differentiable search algorithm DARTS [Liu et al., 2018], where a supernet containing all candidate operators is constructed and architecture parameters are jointly optimized with network parameters through gradient descent. Unfortunately, it has been discussed in SNAS [Xie et al., 2018] that DARTS suffer from the unbounded bias issue towards its objective, which limits the performance of the final derived model. In Tab. 1, we summarize existing AutoGNNs from several perspectives: graph, MPNN space, search algorithm, and task scenario.

## 2.3 GNNs for Link Prediction Task

Even with the great effort invested into the construction and maintenance of networks, many graph data sets still remain incomplete [Schlichtkrull et al., 2018]. Therefore, the link prediction (LP) task is one of the most crucial problems in the graph-structured data, which aims to recover those missing links in a graph [Zhang et al., 2019, 2020a], i.e., predicting the missing node in $e(v_i, ?)$

where ? denotes the target node that has the potential link with $v_i$. Recently, GNN models have been introduced to handle the LP task (abbreviated to GLP models), which can be roughly categorized based on its application scenarios: GLP on homogeneous graphs (i.e., only one type of nodes and edges [Yang et al., 2016]), and multi-relational graphs (i.e., graphs with multiple edge types [Toutanova and Chen, 2015]).

### 2.3.1 Link Prediction on Homogeneous Graphs

As one of the classic approaches for LP task on such homogeneous graphs, heuristic methods predict link existence according to heuristic node similarity scores [Zhang et al., 2020b]. Despite its effectiveness in some scenarios, heuristic methods hold strong assumptions on the link formation mechanism, i.e., highly similar nodes have links. It would fail on those networks where their assumptions do not hold [Zhang and Chen, 2018]. Furthermore, latent feature-based methods [Perozzi et al., 2014, Grover and Leskovec, 2016] factorize some network matrices to learn node embeddings in a transductive way, which limits their generalization ability to unseen data.

Recently, several GLP models have been proposed for the LP task on homogeneous graphs. GAE [Kipf and Welling, 2016b] applies GNN model over the entire graph and aims to learn node embeddings that minimize the graph reconstruction cost through:

$$\mathbf{H} = GCN(\mathbf{X}, \mathbf{A}), \hat{\mathbf{A}} = \sigma(\mathbf{H}\mathbf{H}^\top) \tag{3}$$

where $\mathbf{X} \in \mathbb{R}^{|V| \times D}$ is the feature matrix of nodes, $\mathbf{A} \in \mathbb{R}^{|V| \times |V|}$ is the adjacency matrix, $\mathbf{H}$ is the learned node representations, $\sigma(\cdot)$ is the logistic sigmoid function, $\hat{\mathbf{A}}$ denotes the reconstructed adjacency matrix, whose entry $\hat{\mathbf{A}}_{uv}$ is the predicted score (between 0-1) for target link $e_{uv}$. However, GAE focus on aggregating node attributes only.

SEAL [Zhang and Chen, 2018] and DE-GNN [Li et al., 2020] propose to learn the link embedding from the subgraph structures. Specifically, DE-GNN [Li et al., 2020] considers feature of subgraph structure $\mathbf{X}^{sub}$ for aggregation and utilize node-set level readout:

$$\mathbf{H^{sub}} = GCN(\mathbf{X}^{sub}, \mathbf{A}^{sub}), \ \mathbf{h}_e = R(\{\mathbf{h}_v | v \in S\}), \tag{4}$$

where $S = \{u, v\}$ denotes two nodes in the link $e(u, v)$ for LP task, and $R(\cdot)$ is difference-pooling in DE-GNN, i.e., $R(\cdot) = |\mathbf{h}_u - \mathbf{h}_v|$.

### 2.3.2 Link Prediction on Multi-relational Graphs

Different from graph form $G(E, V)$, the multi-relational graph $G(E, V, R)$ usually contains different types of edges, where $r(u, v)$ indicates the edge type $r \in R$ existing between $u$ and $v$. For example, in recommendation system, users and items are nodes of the bipartite graph, and the edge between nodes could be "click" and "add_to_cart". In the knowledge graph (KG) scenario, nodes represent real-world entities and edges are relations between entities.

Recently, several GLP models have been developed for the LP task on KGs [Schlichtkrull et al., 2018, Vashishth et al., 2019]. Based on the MPNN framework in (1) and (2), R-GCN [Schlichtkrull et al., 2018] proposes to model different edge types through edge-specific weight matrix $\mathbf{W}_r^k$ for $r \in R$, where the MPNN in R-GCN is instantiated as:

$$\mathbf{h}_v^{k+1} = ACT_k\big(\sum\nolimits_{r(u,v)} \mathbf{W}_r^{k+1} \mathbf{h}_u^k \cdot\big) \tag{5}$$

Similar to R-GCN modeling, D-GCN [Marcheggiani and Titov, 2017] and W-GCN [Shang et al., 2019] are also restricted to learning embeddings for nodes only. Instead, CompGCN [Vashishth et al., 2019] propose to jointly learn entity and relation embeddings:

$$\mathbf{h}_v^{k+1} = ACT_k\big(\sum\nolimits_{r(u,v)} \mathbf{W}_{\lambda(r)}^{k+1} \phi(\mathbf{h}_u^k, \mathbf{h}_r^k)\big) \tag{6}$$

where $\mathbf{h}_r^k$ denotes the embedding vector for the specific edge type $r$, and $\lambda(r) \in [incoming, outgoing, self\_loop]$ records information of directed edges. $\phi : \mathbb{R}^d \times \mathbb{R}^d \to \mathbb{R}^d$ can be any entity-relation composition operation, such as $sub$ in TransE [Bordes et al., 2013].

Compared with earlier approaches, GLP models bring remarkable performance gains, illustrating their superiority over LP task. However, exiting GLP models rely on manual and empirical graph neural architecture design, such as selecting proper $ACT(\cdot)$ and $\phi(\cdot)$. Thus, a data-aware GLP model is desired.

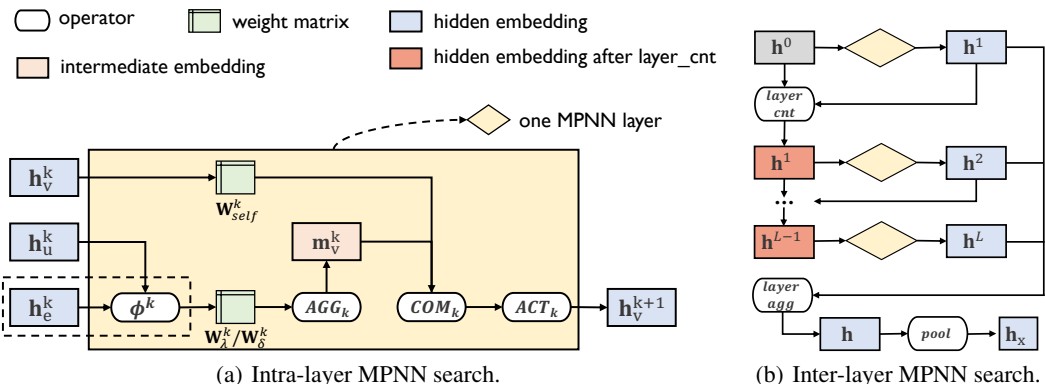

(a) Intra-layer MPNN search.       (b) Inter-layer MPNN search.

Figure 1: The illustration to AutoGEL's search space: a) given representation $\mathbf{h}^k$ in $k$-th layer (including edge embedding $\mathbf{h}_e^k$ if available), AutoGEL searches for proper operators of $\phi^k$, $\mathbf{W}^k$, $AGG_k$, $COM_k$, $ACT_k$ in the intra-layer space, then outputs the hidden node representation $\mathbf{h}_v^{k+1}$. Note that the dotted area will be activated in the scenario of multi-relational graphs.

## 3 AutoGEL: AutoGNN with Explicit Link Information

### 3.1 Search Space

In this subsection, AutoGEL explicitly models the link information in the MPNN space, and proposes several novel operators at both intra-layer (i.e., the message passing in a specific layer) and inter-layer designs (i.e., the message passing across layers). The overflow of space is shown in Fig. 1.

#### 3.1.1 Intra-layer Message Passing Design

As discussed in Sec. 1 and Sec. 2.2, existing AutoGNNs lack modeling of links, which only utilize the pure link information of the node neighborhood. Such implicit way fails to handle and leverage the complex link information. To solve this issue, we present a novel intra-layer message passing framework, which instantiates (1) as:

$$\mathbf{m}_v^{k+1} = AGG_k(\{\mathbf{W}_{\delta(u)}^k \mathbf{h}_u^k : u \in N(v)\}), \tag{7}$$

$$\mathbf{m}_v^{k+1} = AGG_k(\{\mathbf{W}_{\lambda(e)}^k \phi^k(\mathbf{h}_u^k, \mathbf{h}_e^k) : u \in N(v)\}), \tag{8}$$

$$\mathbf{h}_v^{k+1} = ACT_k(COM_k(\{\mathbf{W}_{self}^k \mathbf{h}_v^k, \mathbf{m}_v^{k+1}\})), \tag{9}$$

where (7) and (8) aggregate neighbor information for homogeneous and multi-relational graphs, respectively. $\mathbf{W}_{\delta(u)}^k$ and $\mathbf{W}_{\lambda(u)}^k$ encode parts of the link information as discussed in the following paragraph. For multi-relational graphs, neighboring nodes from different edge types should impose different influence for the center node during message passing. Thus, we utilize the edge embedding $\mathbf{h}_e^k$ in (8) to further encode the type of links, where $\mathbf{h}_e^k$ is updated by $\mathbf{h}_e^{k+1} = \mathbf{W}_{rel}^k \mathbf{h}_e^k$. And we incorporate the composition operator $\phi(\cdot)$ to encode the relationship between edge embedding $\mathbf{h}_e^k$ with node embedding $\mathbf{h}_u^k$. We highlight the differences between the standard MPNN space (see (1) in Sec. 2.2) with (7), (8) and (9) as follows:

- **Linear Transformation** $\mathbf{W}^k$: Given the hidden representation from the last layer, we first apply linear transformation towards embeddings. In (7), we assign neighborhood-type specific weight matrices $\mathbf{W}_{\delta(u)}^k$, where $\delta(u) \in \{self, neigh\}$. $\mathbf{W}_{self}^k$, $\mathbf{W}_{neigh}^k$ are introduced for the node itself and neighbors respectively. This is a weak attention mechanism towards the basic link information for homogeneous graphs, which can distinguish edges between self-type and neighbor-type. In multi-relational graphs, we use edge-aware filters $\mathbf{W}_{\lambda(e)}^k$, where $\lambda(e) \in \{self\_loop, original, inverse\}$ encodes the direction information of edge $e$. We use $\mathbf{W}_{sl}^k$, $\mathbf{W}_O^k$, $\mathbf{W}_I^k$ for self-loop, original, and inverse edge separately.

- **Composition Operator** $\phi^k$ and **Edge Embedding** $\mathbf{h}_e^k$: Following CompGCN [Vashishth et al., 2019], we utilize the composition operator $\phi(\mathbf{h}_u^k, \mathbf{h}_e^k)$ to capture message between the node and edge embeddings before aggregation step. While CompGCN empirically selects the most proper $\phi(\cdot)$ through grid search, we introduce this novel design dimension into our search space, so that

AutoGEL is able to search for the most suitable one together with other design dimensions in a more efficient way. Specifically, we incorporate the candidate operators $\{sub, mult, corr\}$, where $sub(\cdot) = \mathbf{h}_u^k - \mathbf{h}_e^k$ [Bordes et al., 2013], $mult(\cdot) = \mathbf{h}_u^k * \mathbf{h}_e^k$ [Yang et al., 2014], $corr(\cdot) = \mathbf{h}_u^k \star \mathbf{h}_e^k$ [Nickel et al., 2016]. Together with $\mathbf{W}_{\lambda(e)}^k$, the search design on $\phi(\mathbf{h}_u^k, \mathbf{h}_e^k)$ enables AutoGEL to capture semantic meaningful edges by $\mathbf{h}_e^k$, and the interaction between nodes with edges by $\phi(\cdot)$. That is why AutoGEL can handle the LP task on multi-relational graphs, while another edge-level model [You et al., 2020] cannot (see Tab. 1).

### 3.1.2 Inter-Layer Message Passing Design

Traditional MPNNs follows the way in (1), i.e., the input of each MPNN layer is the output of last layer. Motivated by [Xu et al., 2018a, Li et al., 2019], it is beneficial to use the combination of previous layers as input to each layer. In this paper, we also design the inter-layer search space to enables the flexible and powerful GNN architecture to be searched. Specifically, we provide two design dimensions: layer connectivity and layer aggregation.

- **Layer Connectivity:** The literature [Li et al., 2019] have shown that incorporating skip connections (i.e., residual connections and dense connections) across MPNN layers can help alleviating the over-smoothing issue [Li et al., 2018] and empirically improve the model performance. In this work, we conduct systematical investigation towards the joint impact of skip connections together with other design dimension. We select the way of layer connectivity from the set $\{skip, lc\_sum, lc\_concat\}$. Moreover, the layer connectivity function is given as:

$$\mathbf{h}^{k+1} \leftarrow layer\_cnt(\mathbf{h}^k, \mathbf{h}^{k+1}) = \begin{cases} \mathbf{h}^{k+1}, & skip, \\ sum(\mathbf{h}^k, \mathbf{h}^{k+1}), & lc\_sum, \\ \mathbf{W}concat(\mathbf{h}^k, \mathbf{h}^{k+1}), & lc\_concat, \end{cases} \tag{10}$$

  where $\mathbf{h}^k$ denotes embeddings output from $k$-th MPNN intra-layer. As shown in Fig. 1 (b), the representation $\mathbf{h}^k$ will be fed into $k+1$-th MPNN layer to learn $\mathbf{h}^{k+1}$. Then, AutoGEL combines $\mathbf{h}^k$ with $\mathbf{h}^{k+1}$ as in (10) to form a new representation, which will be fed to the next layer. Note that another AutoGNN SANE [Zhao et al.] [2021] does not include $lc\_concat$.

- **Layer Aggregation:** JKNet [Xu et al., 2018a] shows that the layer-wise aggregation allows the adaptive representation learning. The set of candidate layer-wise aggregation defined in AutoGEL is $\{skip, la\_concat, la\_max\}$. Specifically, the layer aggregation function is defined as:

$$\mathbf{h} = layer\_agg(\mathbf{h}^1, \ldots, \mathbf{h}^L) = \begin{cases} \mathbf{h}^L, & skip, \\ [\mathbf{h}^1 || \ldots || \mathbf{h}^L], & la\_concat, \\ max(\mathbf{h}^1, \ldots, \mathbf{h}^L), & la\_max. \end{cases} \tag{11}$$

  Note that $layer\_agg$ aggregates the representations generated from MPNN layers, i.e., those that have not been processed by $layer\_cnt$ (Fig. 1 (b)). And previous AutoGNNs NAS-GCN [Jiang and Balaprakash, 2020] and [You et al., 2020] do not include this operator $layer\_agg$.

### 3.1.3 Pooling

After the intra-layer (Sec. 3.1.1) and inter-layer (Sec. 3.1.2) message passing stages, pooling operation $\mathbf{h}_x = R(\{\mathbf{h}_v | v \in \mathcal{X}\})$ induces high-level representations, where $x$ and $\mathcal{X}$ depend on the given task.

For LP task on homogeneous graphs, the pooling operation outputs the representations of links. In SEAL [Zhang and Chen, 2018], subgraph-level sortpooling method is utilized to readout information from the entire enclosing subgraph for the target link. It is proved in [Srinivasan and Ribeiro, 2019] that joint prediction task only requires joint structure representations of target node-set $S$. Thus, it is not necessary to introduce complex subgraph-level pooling methods. We following DE-GNN [Li et al., 2020] to learn link representations by readout only from target node-set, i.e., $\mathbf{h}_e = R(\{\mathbf{h}_v | v \in S\})$. Note that the original setting difference-pooling for $R(\cdot)$ in DE-GNN does not achieve competitive performance in the empirical study. Instead, we provide the powerful pooling operations $\{sum, max, concat\}$ to be selected for $R(\cdot)$. As for multi-relational graphs, the pooling stage is not required since the edge embedding $\mathbf{h}_e$ is learned in the intra-layer MPNN.

For the node classification task, AutoGEL simply removes $R(\cdot)$ from the search space as literature does. For the graph classification task, the pooling operation outputs high-level graph representation $\mathbf{h}_G = R(\{\mathbf{h}_v | v \in G\})$, and $R(\cdot) \in \{global\_add\_pool, global\_mean\_pool, global\_max\_pool\}$.

## 3.2 Search Algorithm

Given a candidate set $\mathcal{O}$ for a specific operator (e.g., $\{sum, mean, max\}$ for $AGG_k$), let $\mathbf{x}$ be the hidden vector to be fed into this operator, and $\alpha_o$ records the weight that operation $o \in \mathcal{O}$ to be selected. Then the output from this operator is computed as $\bar{o}(\mathbf{x}) = \sum_{o \in \mathcal{O}} \theta_o \cdot o(\mathbf{x})$, where $\theta_o \in \{0, 1\}$. There are multiple operators in AutoGEL's space (see Sec. 3.1), including intra-layer level operators (i.e., $\phi^k, \mathbf{W}^k, AGG_k, COM_k, ACT_k$), inter-layer operators (i.e., $layer\_cnt, layer\_agg$), and pooling operator $R(\cdot)$. Let $\boldsymbol{\theta}$ denote the operation selection for all operators. The GNN search problem can be formulated as $\max_{\boldsymbol{\theta}, \boldsymbol{\omega}} f(\boldsymbol{\theta}, \boldsymbol{\omega}; D)$, where $f(\cdot)$ evaluates the performance of a GNN model $\boldsymbol{\theta}$ with weight $\boldsymbol{\omega}$ on the graph data $D$.

As discussed in Sec. 2.2, the search algorithm adopted in existing AutoGNNs suffers from several issues. Especially, the most similar prior work SANE [Zhao et al., 2021] adopts DARTS [Liu et al., 2018], which directly relaxes $\boldsymbol{\theta}$ to be continuous and makes the objective $f(\boldsymbol{\theta}, \boldsymbol{\omega}; D)$ deterministic differentiable. However, several drawbacks brought by the mixed strategy of DARTS [Liu et al., 2018] have been observed and discussed in the community of neural architecture search. The mixed strategy usually leads to the inconsistent performance issue, i.e., the performance of the derived child network at the end of the searching stage shows significant degradation compared with the performance of the parent network before architecture derivation. That is because the relaxed $\boldsymbol{\theta}$ cannot converge to a one-hot vector [Zela et al., 2019, Chu et al., 2020], thus removing those operations at the end of search actually lead to a different architecture from the final searching result. Moreover, the mixed strategy must maintain all operators in the whole supernet, which requires more computational resources than the one-hot vector [Yao et al., 2020].

Fortunately, SNAS [Xie et al., 2018] leverages the concrete distribution [Maddison et al., 2016, Jang et al., 2016] to propose a stochastic differentiable algorithm, which enables the search objective differentiable with the reparameterization trick. Let a GNN model $\boldsymbol{\theta}$ being sampled from the distribution $p_{\boldsymbol{\alpha}}(\boldsymbol{\theta})$ that parameterized by the structure parameter $\boldsymbol{\alpha}$ as:

$$\theta_o = \frac{\exp((\log \alpha_o - \log(-\log(U_o)))/\tau)}{\sum_{o' \in \mathcal{O}} \exp((\log \alpha_{o'} - \log(-\log(U_{o'})))/\tau)}, \qquad (12)$$

where $\tau$ is the temperature of softmax, and $U_o$ is sampled from the uniform distribution, i.e., $U_o \sim Uniform(0, 1)$. It has been proven that $p(\lim_{\tau \to 0} \theta_o = 1) = \alpha_o / \sum_{o' \in \mathcal{O}} \alpha_{o'}$ [Maddison et al., 2016]. This first guarantees that the probability of $o$ being sampled (i.e., $\theta_o = 1$) is proportional to its weight $\alpha_o$. Besides, the one-hot property $\lim_{\tau \to 0} \theta_o = 1$ makes the stochastic differentiable relaxation unbiased once converged [Xie et al., 2018]. Then the GNN searching problem is reformulated into $\max_{\boldsymbol{\alpha}, \boldsymbol{\omega}} \mathbb{E}_{\boldsymbol{\theta} \sim p_{\boldsymbol{\alpha}}(\boldsymbol{\theta})}[f(\boldsymbol{\theta}, \boldsymbol{\omega}; D)]$, where $\mathbb{E}[\cdot]$ is the expectation. We leverage SNAS [Xie et al., 2018] to optimize the weight of GNN $\boldsymbol{\omega}$ and the weight of operator $\boldsymbol{\alpha}$.

# 4 Experiments

## 4.1 Experimental Setting

AutoGEL[2] is implemented on top of code provided in DE-GNN [Li et al., 2020] and CompGCN [Vashishth et al., 2019] using Pytorch framework [Paszke et al., 2019]. All the experiments are performed using one single RTX 2080 Ti GPU. More details about data sets, hyper-parameter settings and search space designs are introduced in Appendix A.1.1, A.1.2 and A.1.3, respectively.

**Task and Data sets.** For LP task on homogeneous graphs, we follow [Zhang and Chen, 2018, Li et al., 2020] to utilize the datasets: NS [Newman, 2006], Power [Watts and Strogatz, 1998], Router [Spring et al., 2002], C.ele [Watts and Strogatz, 1998], USAir [Batagelj and Mrvar, 2009], Yeast [Von Mering et al., 2002] and PB [Ackland et al., 2005]. As for the LP task on multi-relational graphs, we mainly adopt benchmark knowledge graphs (KGs), FB15k-237 [Toutanova and Chen, 2015] and WN18RR [Dettmers et al., 2018].

---

[2]Code is available at https://github.com/zwangeo/AutoGEL

Table 2: Average AUC (with standard deviation) for LP task on homogeneous graphs

| Type | Model | NS | Power | Router | C.ele | USAir | Yeast | PB |
|---|---|---|---|---|---|---|---|---|
| Heuristic | CN | 94.42±0.95 | 58.80±0.88 | 56.43±0.52 | 85.13±1.61 | 93.80±1.22 | 89.37±0.61 | 92.04±0.35 |
| | RA | 94.45±0.93 | 58.79±0.88 | 56.43±0.51 | 87.49±1.41 | 95.77±0.92 | 89.45±0.62 | 92.46±0.37 |
| | Katz | 94.85±1.10 | 65.39±1.59 | 38.62±1.35 | 86.34±1.89 | 92.88±1.42 | 92.24±0.61 | 92.92±0.35 |
| Latent | SPC | 89.94±2.39 | 91.78±0.61 | 68.79±2.42 | 51.90±2.57 | 74.22±3.11 | 93.25±0.40 | 83.96±0.86 |
| | LINE | 80.63±1.90 | 55.63±1.47 | 67.15±2.10 | 69.21±3.14 | 81.47±10.71 | 87.45±3.33 | 76.95±2.76 |
| | N2V | 91.52±1.28 | 76.22±0.92 | 65.46±0.86 | 84.11±1.27 | 91.44±1.78 | 93.67±0.46 | 85.79±0.78 |
| GLP | VGAE | 94.04±1.64 | 71.20±1.65 | 61.51±1.22 | 81.80±2.18 | 89.28±1.99 | 93.88±0.21 | 90.70±0.53 |
| | PGNN | 94.88±0.77 | - | - | 78.20±0.33 | - | - | 89.72±0.32 |
| | SEAL | 98.85±0.47 | 87.61±1.57 | 96.38±1.45 | 90.30±1.35 | 96.62±0.72 | 97.91±0.52 | 94.72±0.46 |
| | DE-GNN | 99.09±0.79 | 96.68±0.29 | 98.69±0.17 | 89.37±0.17 | 98.04±0.66 | 98.59±0.26 | 94.95±0.37 |
| AutoGNN | AutoGEL | **99.89±0.06** | **98.00±0.21** | **99.08±0.28** | **92.90±1.02** | **98.49±0.45** | **99.24±0.10** | **97.27±0.15** |

For the node classification task, we compare models on three popular citation networks [Sen et al., 2008], i.e., Cora, CiteSeer, and PubMed. For the graph classification task, we adopt four standard benchmarks [Yanardag and Vishwanathan, 2015]: 1) social network datasets: IMDB-BINARY and IMDB-MULTI, 2) bioinformatics datasets: MUTAG and PROTEINS.

**Evaluation Metrics.** For node classification and graph classification, we adopt average accuracy as measurement. We report AUC with standard deviation for LP task on homogeneous graphs. For LP on KGs, we adopt standard evaluation matrices:

- Mean Reciprocal Ranking (MRR): $(\sum_{i=1}^{|S|} 1/rank_i)/|S|$, where $S$ and $rank_i$ denote test triples and ranking results, respectively
- Hits@N: $(\sum_{i=1}^{|S|} \mathbb{1}(rank_i \leq N))/|S|$, where $\mathbb{1}$ denotes indicator function, and $N \in \{1, 3, 10\}$.

**Baselines.** For LP on homogeneous graphs, we use the following approaches as baselines: 1) heuristic methods: CN [Bütün et al., 2018], RA [Zhou et al., 2009], and Katz [Katz, 1953], 2) latent feature based methods: Spectral clustering (SPC) [Tang and Liu, 2011], LINE [Tang et al., 2015] and node2vec (N2V) [Grover and Leskovec, 2016], 3) GLP methods: VGAE [Kipf and Welling, 2016b], PGNN [You et al., 2019], SEAL [Zhang and Chen, 2018], and DE-GNN [Li et al., 2020].

For LP on KGs, we compare AutoGEL with several KG embedding approaches: 1) the geometric models: TransE [Bordes et al., 2013] and RotatE [Sun et al., 2019], 2) bilinear models: DistMult [Yang et al., 2014] and ComplEx [Trouillon et al., 2016], 3) GLP models: R-GCN [Kipf and Welling, 2016b], SACN [Shang et al., 2019], VR-GCN [Ye et al., 2019] and CompGCN [Vashishth et al., 2019], 4) other NN-based models: ConvKB [Nguyen et al., 2017], ConvE [Dettmers et al., 2018], ConvR [Jiang et al., 2019] and HyperER [Balažević et al., 2019].

For the node classification task, we compare AutoGEL with the following baselines: 1) manually designed GNNs: GCN [Kipf and Welling, 2016a], GraphSAGE [Hamilton et al., 2017], GAT [Veličković et al., 2017] and GIN [Xu et al., 2018b], 2) AutoGNNs: GraphNAS [Gao et al., 2020], SANE [Zhao et al., 2021] and [You et al., 2020]. AGNN Zhou et al. [2019] is not included due to no available code.

For the graph classification task, we compare AutoGEL with the following baselines: 1) manually designed GNNs: PATCHY-SAN [Niepert et al., 2016], DGCNN [Zhang et al., 2018], GCN [Kipf and Welling, 2016a], GraphSAGE [Hamilton et al., 2017] and GIN [Xu et al., 2018b], 2) AutoGNNs: [You et al., 2020]. Note that NAS-GCN is specifically designed for molecular property prediction, which is not included in the comparison.

### 4.2 Comparison with GLP models

The model comparison for LP task on homogeneous graphs and knowledge graphs have been summarized in Tab. 2 and 3, respectively. As shown in Tab. 2, heuristic methods perform well on several datasets, but they fail to handle data sets Power and Router. Latent feature-based methods improve the performance on these two data sets but cannot achieve competitive results on other data sets. GLP models outperform heuristic methods and latent feature-based methods, showing their superiority towards LP task. Specifically, DE-GNN is our strongest baseline, where vanilla GCN is adopted to learn node representations for all datasets, then link representation is induced by pooling node embeddings in (4). However, DE-GNN fails to handle the data-diversity issue and

Table 3: MRR and Hits@N for LP task on knowledge graphs

| Type | Model | FB15k-237 | | | | WN18RR | | | |
|---|---|---|---|---|---|---|---|---|---|
| | | MRR | Hits@10 | Hits@3 | Hits@1 | MRR | Hits@10 | Hits@3 | Hits@1 |
| Geometric | TransE | .294 | .465 | - | - | .226 | .501 | - | - |
| | RotatE | .338 | .533 | .375 | .241 | .476 | **.571** | .492 | .428 |
| Bilinear | DisMult | .241 | .419 | .263 | .155 | .430 | .490 | .440 | .390 |
| | ComplEx | .247 | .428 | .275 | .158 | .440 | .510 | .460 | .410 |
| NN-based | ConvKB | .243 | .421 | .371 | .155 | .249 | .524 | .417 | .057 |
| | ConvE | .325 | .501 | .356 | .237 | .430 | .520 | .440 | .400 |
| | ConvR | .350 | .528 | .385 | .261 | .475 | .537 | .489 | .443 |
| | HyperER | .341 | .520 | .376 | .252 | .465 | .522 | .477 | .436 |
| GLP | R-GCN | .248 | .417 | - | .151 | - | - | - | - |
| | SACN | .350 | **.540** | .390 | .260 | .470 | .540 | .480 | .430 |
| | VR-GCN | .248 | .432 | .272 | .159 | - | - | - | - |
| | CompGCN | .355 | .535 | .390 | .264 | **.479** | .546 | **.494** | .443 |
| AutoGNN | AutoGEL | **.357** | .538 | **.391** | **.266** | **.479** | .549 | .492 | **.444** |

Table 4: Average accuracy (%) for node classification and graph classification

| Type | Model | Node Classification | | | Graph Classification | | | |
|---|---|---|---|---|---|---|---|---|
| | | Cora | CiteSeer | Pubmed | IMDB-B | IMDB-M | MUTAG | PROTEINS |
| Manual GNNs | PATCHYSAN | - | - | - | 71.00 | 45.20 | 92.60 | 75.90 |
| | DGCNN | - | - | - | 70.00 | 47.80 | 85.80 | 75.50 |
| | GCN | 88.11 | 76.66 | 88.58 | 74.00 | 51.90 | 85.60 | 76.00 |
| | GraphSAGE | 87.41 | 75.99 | 88.34 | 72.30 | 50.90 | 85.10 | 75.90 |
| | GAT | 87.19 | 75.18 | 85.73 | - | - | - | - |
| | GIN | 86.00 | 73.40 | 87.99 | 75.10 | 52.30 | 89.40 | 76.20 |
| AutoGNN | GraphNAS | 88.40 | 77.62 | 88.96 | - | - | - | - |
| | SANE | 89.26 | **78.59** | 90.47 | - | - | - | - |
| | You et al. [2020] | 88.50 | 74.90 | - | - | 47.80 | - | 73.90 |
| | AutoGEL | **89.89** | 77.66 | 89.68 | **81.20** | **56.80** | **94.74** | **82.68** |

cannot consistently achieve leading performance on all data sets. In this paper, AutoGEL first adopts the pooling way in DE-GNN, then enables a more flexible way to select the most suitable pooling function $R(\cdot)$ (see Sec. 3.1.3) instead of the fixed pooling function in DE-GNN. Searching the pooling function and other operators make AutoGEL handle the data-diversity issue, and consistently achieve the state-of-the-art performance for LP task on homogeneous graphs. Furthermore, we demonstrate the model performance of LP task on knowledge graphs in Tab. 3. Note that the improvements on KGs is not as obvious as that on homogeneous graphs. In practice, GLP models run longer than Geometric and Bilinear models, which leads to the difficulty of tuning hyper-parameters (see more discussions in Appendix A.4).

Moreover, we present several cases of searched architectures in Appendix A.2. And we show several ablation studies to provide some insights into the AutoGEL space design in Appendix A.3, including the impacts of the inter-level design, pooling operator, weight transformation matrices, and edge embedding.

### 4.3 Comparison with AutoGNN models

To compare with other AutoGNNs, we also demonstrate the performance of AutoGEL on node-level and graph-level tasks. The empirical comparison on the node classification and graph classification task is shown in Tab. 4. AutoGEL shows the great generalization ability towards different graph tasks. AutoGEL outperforms all the manually designed GNN baselines and also achieves competitive results with existing AutoGNNs designed specifically for these tasks. The data sets for node classification task usually contains rich node features. AutoGEL simplifies the attention mechanism in existing GNNs for node classification from $a_{uv}$ (see Sec. 2.2) to $\mathbf{W}_{\delta(u)}^k$ in (7), which leads to slightly inadequate performance. We notice that AutoGEL brings more significant performance gains on the graph classification task. The data sets for graph classification have not sufficient node features

Table 5: The search time (clock time in seconds) of AutoGNNs on the node classification task.

| Model | Cora | Citeseer | PubMed |
|---|---|---|---|
| GraphNAS [Gao et al., 2020] | 3240 | 3665 | 5917 |
| SANE [Zhao et al., 2021] | 14 | 35 | 54 |
| AutoGEL | 12 | 16 | 19 |

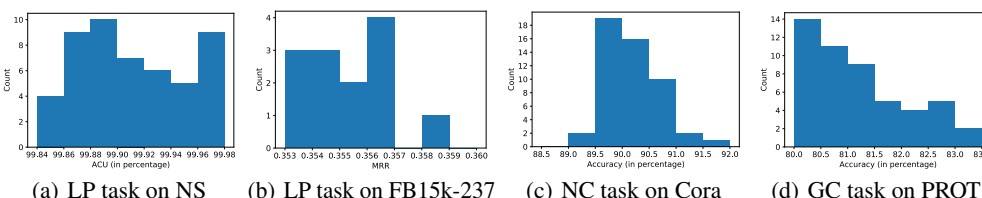

(a) LP task on NS    (b) LP task on FB15k-237    (c) NC task on Cora    (d) GC task on PROT

Figure 2: Performance frequency statistics over multiple runs for each task

as those data sets for node classification, which requires effective learning from graph structures. AutoGEL is more suitable for this task by learning from edges.

We also compare the search efficiency between AutoGNNs in Tab. 5. Statistics for other AutoGNNs are taken from the start-of-the-art SANE [Zhao et al., 2021], which sets search epochs to 200 for all the AutoGNN baselines. To reduce search cost in GraphNAS, SANE and AutoGEL leverages the idea of parameter sharing [Pham et al., 2018] to avoid repeatedly training weights of different sampled GNN architectures. Moreover, AutoGEL adopts a more advanced search algorithm compared with SANE (see Sec. 3.2), thereby further reduces the search cost. We show experimental results of a variant of AutoGEL in search algorithm in Appendix A.3. And more details about search efficiency are reported in Appendix A.4.

### 4.4 The Empirical Study on Robustness

All effectiveness results in the main context (Tab. 2, Tab. 3, and Tab. 4) are reported under the average of 4 runs. Note that Tab. 3 and Tab. 4 do not contain variance due to space limits. To illustrate the robustness of AutoGEL, here we report results after multiple runs of AutoGEL on several tasks in Fig. 2. We set the number of different runs as 10 for FB15k-237 dataset and 50 for the rest, due to the relative longer running time required for FB15k-237. We can see that even in the some worst cases, AutoGEL still rival or surpass its strongest baselines over all the tasks, its indicating the effectiveness.

## 5 Conclusion

In this paper, we present a novel AutoGNN with explicit link information, named AutoGEL. Specifically, AutoGEL incorporates the edge embedding in the MPNN space, and proposes several novel design dimensions at intra-layer and inter-layer designs. Moreover, AutoGEL upgrades the search algorithm of AutoGNNs by leveraging a promising NAS algorithm SNAS. Experimental results well demonstrate that AutoGEL not only achieves the leading performance on the LP task, but also shows competitive results on the node and graph classification tasks.

For future works, one direction worth trying is to adapt AutoGNNs to the LP task on hyper-relational KGs, which contain a lot of hyper-relational facts $r(u_1, \ldots, u_n), n \geq 2$. First, the pioneer data-aware methods for LP tasks on KGs are mainly based on the bilinear models, such as AutoSF [Zhang et al., 2020c] and ERAS [Shimin et al., 2021]. Introducing the MPNN space can promote a more comprehensive search space because the composition operator is not limited to the bilinear models. Second, using the multi-relational hypergraph could be a more natural way to model hyper-relational facts [Yadati, 2020, Di et al., 2021]. Another interesting direction is to search GNN architectures on dynamic graph data sets. Note that search efficiency would be the most challenging issue. One of the key points is to make full use of previous well-trained GNN controllers.

# 6 Acknowledgements

Lei Chen's work is partially supported by National Key Research and Development Program of China Grant No. 2018AAA0101100, the Hong Kong RGC GRF Project 16202218, CRF Project C6030-18G, C1031-18G, C5026-18G, RIF Project R6020-19, AOE Project AoE/E-603/18, Theme-based project TRS T41-603/20R, China NSFC No. 61729201, Guangdong Basic and Applied Basic Research Foundation 2019B151530001, Hong Kong ITC ITF grants ITS/044/18FX and ITS/470/18FX, Microsoft Research Asia Collaborative Research Grant, HKUST-NAVER/LINE AI Lab, Didi-HKUST joint research lab, HKUST-Webank joint research lab grants.

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
