Table 6: Dataset statistics for link prediction task

| | Homogeneous Graphs | | | | | | | KGs | |
| --- | --- | --- | --- | --- | --- | --- | --- | --- | --- |
| | NS | Power | Router | C.ele | USAir | Yeast | PB | FB15k237 | WN18RR |
| # Nodes | 1589 | 4941 | 5022 | 297 | 332 | 2375 | 1222 | 14541 | 40943 |
| # Edges | 2742 | 6594 | 6258 | 2148 | 2126 | 11693 | 16714 | 310116 | 93003 |
| # Edge types | 1 | 1 | 1 | 1 | 1 | 1 | 1 | 237 | 11 |
| # Relations | - | - | - | - | - | - | - | 237 | 11 |
| Avg. # Degrees | 3.45 | 2.67 | 2.49 | 14.46 | 12.81 | 9.85 | 27.36 | 21.33 | 2.27 |
| # Training | 4387 | 10550 | 10012 | 3436 | 3401 | 18708 | 26742 | 272,115 | 86,835 |
| # Validation | 548 | 1319 | 1251 | 429 | 425 | 2338 | 3342 | 17,535 | 3,034 |
| # Testing | 548 | 1319 | 1251 | 429 | 425 | 2338 | 3342 | 20,466 | 3,134 |

Table 7: Dataset statistics for node classification and graph classification task

| | Node Classification | | | Graph Classification | | | |
| --- | --- | --- | --- | --- | --- | --- | --- |
| | Cora | CiteSeer | PubMed | IMDB-B | IMDB-M | MUTAG | PROTEINS |
| # Graphs | 1 | 1 | 1 | 1000 | 1500 | 188 | 1113 |
| # Nodes | 2708 | 3327 | 19717 | 19.8 (Avg.) | 13.0 (Avg.) | 17.9 (Avg.) | 39.1 (Avg.) |
| # Edges | 2742 | 6594 | 6258 | 96.53 (Avg.) | 65.94 (Avg.) | 19.79 (Avg.) | 72.82 (Avg.) |
| # Edge types | 1 | 1 | 1 | 1 | 1 | 4 | 1 |
| # Node Attr. | 1433 | 3703 | 500 | - | - | 3 | 7 |
| # Classes | 7 | 6 | 3 | 2 | 3 | 2 | 2 |

# A  Experiments

## A.1  More Experimental Settings

### A.1.1  Dataset Details

We summarize the dataset statistics in Tab. 6 and Tab. 7. In terms of dataset splits, for LP task on homogeneous graphs, we follow [Li et al., 2020] to split 80%, 10%, 10% of existing links for training, validation and testing respectively. The same number of negative links are also included through random sampling. During training phase, positive test links are removed to avoid label leakage. For KGs, we follow their standard split as shown in Tab. 6. Moreover, we use 60%, 20%, 20% dataset split for node classification as in [Zhao et al., 2021], and 80%, 10%, 10% for graph classification task to keep the same percentage of test split as in [Xu et al., 2018b] for fair comparison.

### A.1.2  Hyperparameter Settings

We provide detailed hyperparameter settings in Tab. 8 for our implementation. Hyperparameters are tuned through hyperopt [3] Bergstra et al. [2013].

Table 8: List of value / range of hyperparameters in AutoGEL's implementation

| Hyperparameters | Link Prediction | | Node Classification | Graph Classification |
| --- | --- | --- | --- | --- |
| | Homo. Graphs | KGs | | |
| Optimizer | Adam | Adam | Adam | Adam |
| Learning rate | 1e-4 | {1e-3,1e-4} | {1e-3, 5e-3, 1e-4} | {1e-2, 1e-3, 1e-4} |
| MPNN layers | 2 | {1, 2} | 2 | 4 |
| Batch size | {64, 128} | {128, 256} | {64, 256} | {32, 128} |
| Hidden dimension | 100 | 200 | {64, 256} | {16, 32, 64} |
| Dropout | {0, 0.2} | {0, 0.1, 0.2, 0.3} | {0, 0.5} | {0, 0.5} |
| Search epoch | 300 | {200,300} | {30, 200} | {30, 200} |

---

[3]https://github.com/hyperopt/hyperopt

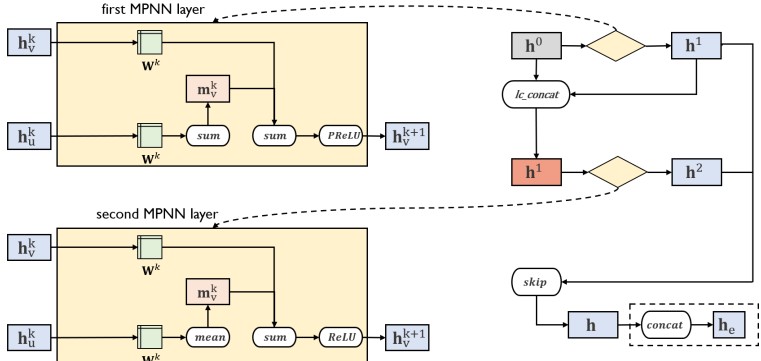

Figure 3: An example: GNN architecture searched by AutoGEL for LP task on PB dataset.

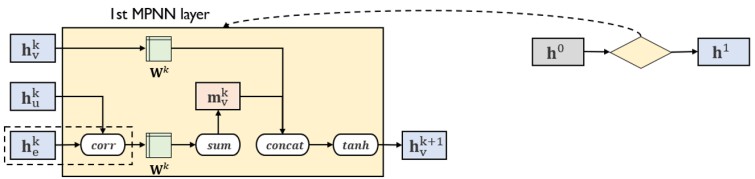

Figure 4: An example: GNN architecture searched by AutoGEL for LP task on FB15k-237 dataset.

### A.1.3 Search Space

Apart from our main designs presented in Section 3.1, AutoGEL also includes several other intra-level design dimensions in the search space:

- **Aggregation** $AGG_k$: We follow the common design in AutoGNN works (please refer to Section 2.2 for more details) to include $\{sum, mean, max\}$ for neighborhood aggregation.

- **Combination** $COM_k$: We select combination function from $\{sum, concat\}$. We omit $mlp$ combination since we empirically find simpler combination operator $sum$ and $concat$ adopted in our search space already achieves good performance.

- **Activation** $ACT_k$: Empirical observations from [You et al., 2020] shows the superiority of PReLU as the activation function for GNNs. In this work, we restrict our candidate activation functions in $\{ReLU, PReLU\}$. For the LP task on KGs, we follow the alternative setting to use $tanh$ since we empirically found $ReLU$ and $PReLU$ not suitable.

- **Node Labeling**: The node labeling method (e.g., double-radius node labeling (DRNL) [Zhang and Chen, 2018] and distance encoding (DE) [Li et al., 2020]) is an important component towards the success of structural prediction tasks (e.g., link prediction). AutoGEL presets the DE as the node labeling approach for the LP task due to its generality and empirically good performance. DRNL can be regarded as a special case for DE, where the differences between them are marginal. Both DE and DRNL work well in practice [Li et al., 2020]. Moreover, we tried to incorporate this design dimension into the search space and enable it to be jointly searched with other architecture components. Unfortunately, sacrificing some search efficiency may not be able to improve the effectiveness because DE is already a powerful technique. Out of this concern, AutoGEL presets DE as the node labeling method to better balance between effectiveness and efficiency.

### A.2 Case Study

Here we show some searched architectures for several tasks: link prediction (LP), node classification (NC), and graph classification (GC).

For the LP task (see Fig. 3 and Fig. 4), we find that the depth of MPNN layers $L$ leading to highest performance is different from graph scenarios. Generally, $L = 2$ for homogeneous graphs while $L = 1$ for knowledge graph. One possible reason is that KGs are usually more densely connected (see Tab. 6 for more dataset details), and deeper MPNN layers would cause the over-smoothing issue,

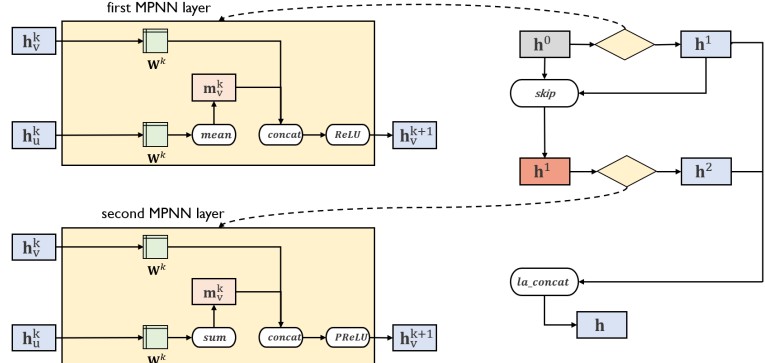

Figure 5: An example: GNN architecture searched by AutoGEL for the NC task on PubMed.

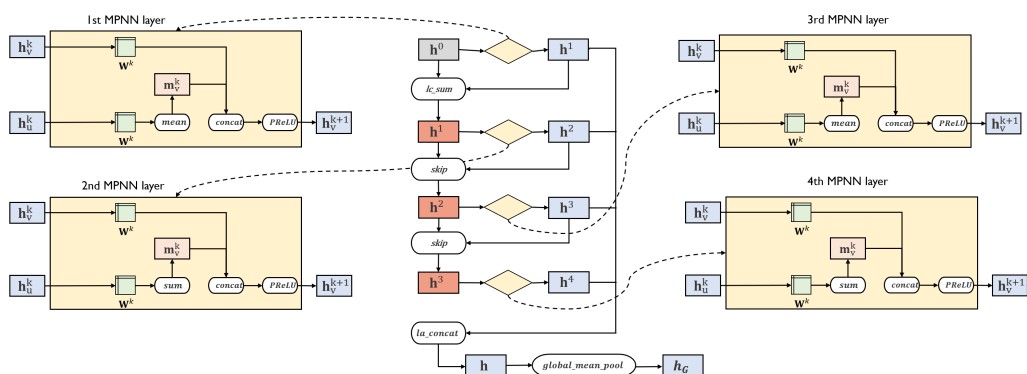

Figure 6: An example: GNN architecture searched by AutoGEL for the GC task on PROTEINS

resulting in performance degradation. Moreover, it is also discussed in [Zhang and Chen, 2018] that for subgraph-based LP approaches on homogeneous graphs adopted by AutoGEL, 2-hop enclosing subgraphs already contain rich information required for the prediction, therefore $L > 2$ should not be very necessary.

Specifically, for the LP task on KG scenario, we empirically observe that the composition operator $\phi(\mathbf{h}_u, \mathbf{h}_e)$ (see Sec. 3.1) should be one of the most critical components. This operator determines the way how to compose the neighborhood embedding $\mathbf{h}_u$ and edge embedding $\mathbf{h}_e$ to generate the message for the center node $v$. Actually, the composition operator $\phi$ incorporates the scoring function design in past KG embedding models, such as subtraction for geometric models and multiplication for bilinear models. From experiments, we observed that $\phi$ is data-dependent. $corr$ is more preferred for the FB15k-237 dataset, and simpler $mult$ is prone to get better results for the WN18RR dataset. Using others $\phi$ for these data sets would lead to significantly different performance based on the empirical study.

For the NC task (see Fig. 5), pooling operator $R(\cdot)$ is removed from the search space, and we set $L = 2$ for all three citation datasets, since we observe performance degradation with larger $L$ on those datasets.

For the GC task (see Fig. 6), we empirically observe that AutoGEL prefers deeper GNN architectures compared to the LP and NC tasks. One potential reason is that, citation datasets adopted for the NC task are similar to "small world" networks [Barthélémy and Amaral, 2011] where each node can reach the entire graph within just a few hops. But the data sets for the GC task represent graph structures, such as molecules, where deeper architectures might be beneficial to increase effective receptive field. Besides, while the NC task mainly relies on local neighborhood (short-range) information, the GC task may require long-range information to capture certain graph properties that are essential to the prediction, such as chemical properties of molecules [Matlock et al., 2019], and graph moments [Dehmamy et al., 2019]. Thus deeper GNN architectures are more desired.

Table 9: Average AUC (with standard deviation) for LP task on homogeneous graphs

| Type | Model | NS | Power | Router | C.ele | USAir | Yeast | PB |
|---|---|---|---|---|---|---|---|---|
| Heuristic | CN | 94.42±0.95 | 58.80±0.88 | 56.43±0.52 | 85.13±1.61 | 93.80±1.22 | 89.37±0.61 | 92.04±0.35 |
| | RA | 94.45±0.93 | 58.79±0.88 | 56.43±0.51 | 87.49±1.41 | 95.77±0.92 | 89.45±0.62 | 92.46±0.37 |
| | Katz | 94.85±1.10 | 65.39±1.59 | 38.62±1.35 | 86.34±1.89 | 92.88±1.42 | 92.24±0.61 | 92.92±0.35 |
| Latent | SPC | 89.94±2.39 | 91.78±0.61 | 68.79±2.42 | 51.90±2.57 | 74.22±3.11 | 93.25±0.40 | 83.96±0.86 |
| | LINE | 80.63±1.90 | 55.63±1.47 | 67.15±2.10 | 69.21±3.14 | 81.47±10.71 | 87.45±3.33 | 76.95±2.76 |
| | N2V | 91.52±1.28 | 76.22±0.92 | 65.46±0.86 | 84.11±1.27 | 91.44±1.78 | 93.67±0.46 | 85.79±0.78 |
| GLP | VGAE | 94.04±1.64 | 71.20±1.65 | 61.51±1.22 | 81.80±2.18 | 89.28±1.99 | 93.88±0.21 | 90.70±0.53 |
| | PGNN | 94.88±0.77 | - | - | 78.20±0.33 | - | - | 89.72±0.32 |
| | SEAL | 98.85±0.47 | 87.61±1.57 | 96.38±1.45 | 90.30±1.35 | 96.62±0.72 | 97.91±0.52 | 94.72±0.46 |
| | DE-GNN | 99.09±0.79 | 96.68±0.29 | 98.69±0.17 | 89.37±0.17 | 98.04±0.66 | 98.59±0.26 | 94.95±0.37 |
| AutoGNN | AutoGEL | **99.89±0.06** | **98.00±0.21** | **99.08±0.28** | **92.90±1.02** | **98.49±0.45** | **99.24±0.10** | **97.27±0.15** |
| | AutoGEL-intra | 99.85±0.06 | 97.65±0.21 | 98.92±0.23 | 92.36±1.13 | 98.29±0.49 | 99.18±0.09 | 97.16±0.13 |
| | AutoGEL-diff | 99.58±0.17 | 97.05±0.19 | 98.92±0.27 | 90.38±0.64 | 97.89±0.69 | 98.90±0.10 | 96.12±0.21 |
| | AutoGEL-\δ | 99.85±0.06 | 97.65±0.20 | 98.98±0.23 | 92.58±1.14 | 98.33±0.39 | 99.14±0.09 | 97.23±0.07 |
| | AutoGEL-darts | 99.85±0.06 | 97.31±0.09 | 98.87±0.23 | 91.98±0.77 | 97.98±0.42 | 99.02±0.13 | 95.84±0.29 |

Table 10: MRR and Hits@N for LP task on knowledge graphs

| Type | Model | FB15k-237 | | | | WN18RR | | | |
|---|---|---|---|---|---|---|---|---|---|
| | | MRR | Hits@10 | Hits@3 | Hits@1 | MRR | Hits@10 | Hits@3 | Hits@1 |
| Geometric | TransE | .294 | .465 | - | - | .226 | .501 | - | - |
| | RotatE | .338 | .533 | .375 | .241 | .476 | **.571** | .492 | .428 |
| Bilinear | DisMult | .241 | .419 | .263 | .155 | .430 | .490 | .440 | .390 |
| | ComplEx | .247 | .428 | .275 | .158 | .440 | .510 | .460 | .410 |
| NN-based | ConvKB | .243 | .421 | .371 | .155 | .249 | .524 | .417 | .057 |
| | ConvE | .325 | .501 | .356 | .237 | .430 | .520 | .440 | .400 |
| | ConvR | .350 | .528 | .385 | .261 | .475 | .537 | .489 | .443 |
| | HyperER | .341 | .520 | .376 | .252 | .465 | .522 | .477 | .436 |
| GLP | R-GCN | .248 | .417 | - | .151 | - | - | - | - |
| | SACN | .350 | **.540** | .390 | .260 | .470 | .540 | .480 | .430 |
| | VR-GCN | .248 | .432 | .272 | .159 | - | - | - | - |
| | CompGCN | .355 | .535 | .390 | .264 | **.479** | .546 | **.494** | .443 |
| AutoGNN | AutoGEL | **.357** | .538 | **.391** | **.266** | **.479** | .549 | .492 | **.444** |
| | AutoGEL-\λ | .355 | .533 | .389 | .265 | .470 | .532 | .486 | .434 |
| | AutoGEL-darts | .356 | .538 | **.391** | .265 | .472 | .544 | .485 | .434 |
| | AutoGEL-\h$_e$ | .355 | .531 | .389 | .265 | .454 | .540 | .483 | .402 |

## A.3 Ablation Study

Apart from the main experiment results shown in Sec. 4, we also conduct several ablation studies to investigate the influence of different components in AutoGEL and provide additional experiments in this section.

*1) Impact of Inter-level Design:* AutoGEL provides various design dimensions from both intra-level (see Sec. 3.1.1) as well as inter-level (see Sec. 3.1.2). To study the effect of the proposed inter-level designs, we set a variant, i.e., AutoGEL-intra, where inter-level design dimensions are removed from the search space, and we only conduct operator search from intra-level. As shown in Tab. 9 and Tab. 11, AutoGEL-intra achieves competitive performance compared with manually-designed GNN baselines, which illustrates the powerfulness of AutoGEL's intra-level designs. But AutoGEL brings more performance gains over AutoGEL-intra by searching inter-level operators. Note that the number of layers $L$ for the LP task on KG is usually 1 (see Appendix A.2), thus there are no results of AutoGEL-intra in Tab. 10.

*2) Impact of Pooling Operator:* In this paper, we provide pooling operation candidates $R(\cdot) \in \{sum, mean, max\}$ for the LP task on homogeneous graphs. As discussed in Sec. 3.1.3, DEGNN [Li et al., 2020] utilizes the difference-pooling as $R(\cdot)$. Here we set a variant, i.e., AutoGEL-diff, where we remove the proposed pooling candidates from the search space and fix the difference-pooling instead. As shown in Tab. 9, the fixed difference-pooling method leads to the significant performance degradation, illustrating the strength of AutoGEL's pooling design.

Table 11: Average accuracy (%) for node classification and graph classification

| Type | Model | Node Classification | | | Graph Classification | | | |
|------|-------|------|---------|--------|--------|--------|-------|----------|
| | | Cora | CiteSeer | Pubmed | IMDB-B | IMDB-M | MUTAG | PROTEINS |
| Manual GNNs | PATCHYSAN | - | - | - | 71.00 | 45.20 | 92.60 | 75.90 |
| | DGCNN | - | - | - | 70.00 | 47.80 | 85.80 | 75.50 |
| | GCN | 88.11 | 76.66 | 88.58 | 74.00 | 51.90 | 85.60 | 76.00 |
| | GraphSAGE | 87.41 | 75.99 | 88.34 | 72.30 | 50.90 | 85.10 | 75.90 |
| | GAT | 87.19 | 75.18 | 85.73 | - | - | - | - |
| | GIN | 86.00 | 73.40 | 87.99 | 75.10 | 52.30 | 89.40 | 76.20 |
| AutoGNN | GraphNAS | 88.40 | 77.62 | 88.96 | - | - | - | - |
| | SANE | 89.26 | 78.59 | 90.47 | - | - | - | - |
| | You et al. [2020] | 88.50 | 74.90 | - | - | 47.80 | - | 73.90 |
| | AutoGEL | 89.66 | 77.66 | 90.00 | 81.20 | 56.80 | 96.14 | 82.68 |
| | AutoGEL-intra | 88.93 | 76.33 | 89.73 | 77.62 | 55.58 | 95.98 | 77.96 |
| | AutoGEL-\$\delta$ | 88.88 | 76.55 | 89.96 | 77.44 | 53.88 | 93.75 | 79.3 |
| | AutoGEL-darts | 89.00 | 77.49 | 89.85 | 76.69 | 47.42 | 96.05 | 80.08 |

Table 12: Search time (clock time in seconds) comparison on the node classification (NC) task and graph classification (GC) task

| | Node Classification | | | Graph Classification | | | |
|---|------|----------|--------|--------|--------|-------|----------|
| | Cora | CiteSeer | PubMed | IMDB-B | IMDB-M | MUTAG | PROTEINS |
| AutoGEL | 12 | 16 | 19 | 58 | 90 | 2.4 | 56 |
| AutoGEL-darts | 15 | 31 | 97 | 122 | 138 | 3.8 | 95 |

Table 13: Search time (clock time in hours) comparison on the LP task

| | NS | Power | Router | C.ele | USAir | Yeast | PB | FB15k-237 | WN18RR |
|---|-----|-------|--------|-------|-------|-------|------|-----------|--------|
| AutoGEL | 0.5 | 2.6 | 3.4 | 1.4 | 1.3 | 4.0 | 14.4 | 18.1 | 13.1 |
| AutoGEL-darts | 1.0 | 2.7 | 3.4 | 1.5 | 1.4 | 6.0 | 14.7 | 18.3 | 13.7 |

*3) Impact of Separate Weight Transformation Matrices:* AutoGEL provide novel linear transformation approaches, i.e., we assign neighborhood-type specific matrices $\mathbf{W}^k_{\delta(u)}$ as special attention mechanism for homogeneous graphs, and edge-aware filters $\mathbf{W}^k_{\lambda(e)}$ to incorporate information from different directions for heterogeneous graphs (see Sec. 3.1.1). To study the impact of such designs, we provide two variants, i.e., AutoGEL-\$\delta$ (see Tab. 9 and 11) and AutoGEL-\$\lambda$ (see Tab. 10), where $\mathbf{W}^k_{\delta(u)}$ and $\mathbf{W}^k_{\lambda(e)}$ are simply replaced by a single $\mathbf{W}^k$. Compared with these two variants, AutoGEL achieves better performance cross different graph tasks and datasets.

*4) Impact of Edge Embedding:* To show the effectiveness of edge embedding $\mathbf{h}_e$ on the LP task, we set another variant, i.e., AutoGEL-\$\mathbf{h}_e$, by removing $\mathbf{h}_e$ from AutoGEL's MPNN and simply replace $\phi(\mathbf{h}^k_u, \mathbf{h}^k_e)$ with $\mathbf{h}^k_u$ in (8). Experiment results are shown in Tab. 10. Performance degradation is observed for the AutoGEL-\$\mathbf{h}_e$, especially on the WN18RR dataset, indicating that $\mathbf{h}_e$ is indeed a critical design.

*5) Impact of Stochastic Differentiable Search Algorithm:* As discussed in Sec.3.2, AutoGEL adopts stochastic differentiable search algorithm in SNAS to perform more effective and efficient architecture search. To show its superiority, we also try the deterministic differentiable search algorithm DARTS for AutoGEL, denoted as AutoGEL-darts. Tab. 9, Tab. 10, and Tab. 11 empirically show the consistent superior performance of the AutoGEL compared with AutoGEL-darts variant cross node/edge/graph level tasks, indicating the effectiveness of AutoGEL's search algorithm. Besides, we further show that the search cost of AutoGEL is also lower than its AutoGEL-darts variant (see Tab. 12, and Tab. 13).

Table 14: Running time (clock time in hours) of AutoGEL and several baselines for the LP task

| Type | Model | HGs | | | | | | | KGs | |
|---|---|---|---|---|---|---|---|---|---|---|
| | | NS | Power | Router | C.ele | USAir | Yeast | PB | FB15k-237 | WN18RR |
| GLP for HG | DE-GNN | 0.1 | 1.0 | 1.2 | 0.2 | 0.3 | 2.0 | 4.7 | - | - |
| Bilinear for KG | DistMult | - | - | - | - | - | - | - | 2.6 | 0.4 |
| NN for KG | ConvE | - | - | - | - | - | - | - | 26.0 | 10.2 |
| GLP for KG | CompGCN | - | - | - | - | - | - | - | 16.1 | 7.8 |
| Ours | AutoGEL (search) | 0.5 | 2.6 | 3.4 | 1.4 | 1.3 | 4.0 | 14.4 | 18.1 | 13.1 |
| | AutoGEL (training) | 0.3 | 0.5 | 0.7 | 0.4 | 0.4 | 1.5 | 4.8 | 13.1 | 7.3 |

## A.4 Search Efficiency

As mentioned in Sec. 4.2, we found that existing GLP modes generally require more computational resources in practice. Thus, we try to reduce the search cost in the proposed AutoGEL. Tab. 14 reports the running time (hours) of AutoGEL and several other representative baselines for the LP task on the homogenous graph (HG) and knowledge graph (KG).

From Tab. 14, we can observe that: On the LP task on HGs (NA, Power, Router, C.ele, USAir, Yeast, and PB), AutoGEL runs quite fast, which substantially eases the difficulty of using AutoGEL. Besides, AutoGEL achieves more significant performance boost in this scenario (see Tab. 2). On the LP task on KGs (FB15K-237, WN18RR), DistMult [Yang et al., 2014] is a representative of bilinear models that run much faster among all KGE models. Although AutoGEL is slower than DistMult, its computational cost is very close with classic neural networks (NNs) for KG ConvE [Dettmers et al., 2018] and GLP model CompGCN [Vashishth et al., 2019]. Then recalling Tab. 3, AutoGEL well balances between search cost and effectiveness.