# OpenReview forum: "AutoGEL: An Automated Graph Neural Network with Explicit Link Information"
_NeurIPS.cc/2021/Conference — NeurIPS 2021 Poster_

### Official Review · Reviewer_GtXj · 2021-07-08

**Rating:** 6
**Confidence:** 5

**Summary:**

This paper proposes a neural architecture search method for using GNNs for link prediction. The proposed method explicitly models edge information in the search process, and proposes several novel design and search space choices for AutoGNN. Through a DARTS-like differentiable search method, the proposed method achieves state-of-the-art performance on various link prediction and node/graph classification tasks.

**Limitations And Societal Impact:**

No. One concern is the search efficiency (in general) for NAS methods. The paper discusses this issue in Section 4.3. Another concern is that the search space does not incorporate different node labeling methods for GNN link prediction, such as distance encoding, which are shown to be crucial for success of GNNs on link prediction. Thirdly, I suggest demonstrating one or two common findings from the searched architectures: what components of a GNN are most important for link prediction?

**Main Review:**

Originality: Link prediction on graphs is an important task yet receives less attention than node/graph classification. This paper extends AutoGNN to link prediction and obtains SoTA results, which is highly appreciated for the community. The related work is sufficient, especially for including some latest arxiv papers on GNN link prediction.

Quality: The techincal part of the paper is ok. It is an application of an existing NAS algorithm SNAS with some novel design of search spaces. Nevertheless, the proposed search space choices are novel and reasonable to me.

Clarity: The paper is easy-to-understand and written in an intuitive way.

Significance: Through extensive experimental results, the proposed AutoGNN method demonstrates SoTA results across a number of tasks (not restricted to link prediction), outperforming manually designed models. This demonstrates the importance for explicit modeling edge information and link prediction in designing GNN architectures.

**Time Spent Reviewing:**

1.5

---

> ### Author Response · Authors · 2021-08-10
> **We replied to several concerns raised by the reviewer GtXj, including more discussions and more experimental reports.**
>
> We sincerely thank the reviewer's efforts in reviewing our submission, and our detailed replies are as follows:
>
>
> ## Replies to Detailed Questions
>
> **RGtXj-D1:** One concern is the search efficiency for neural architecture search (NAS) methods mentioned by the authors.
>
>
> **Reply to RGtXj-D1:**
> Yes, it is inevitable that NAS methods generally take longer time than manually designed ones. But generally, AutoGEL is more efficient than other AutoGNNs by introducing a compressed yet representative search space and leveraging a fast and robust search algorithm.
> First, the proposed search space enables the powerful GNN model to be searched, but the size of the search space is not too large to increase the search cost.
> Second, the adopted search algorithm follows the one-shot way to search architectures, which is faster than stand-alone NAS search algorithms [1].
>
> In the submission,
> we demonstrated the search efficiency for AutoGEL in terms of several graph tasks in Tab. 5, 9, 10.
> As pointed out by other reviewers, we add more efficiency comparison in the responses (see Rq5RS-D4 and RK8wz-D3).
> Here we summarized more reports about efficiency as below:
>
> - AutoGEL adopted a fast NAS search algorithm
>
> | (seconds) | Cora | CiteSeer | PubMed |
> | :----: | :----: | :----: | :----: |
> | AutoGEL | 12 | 16 | 19 |
> | AutoGEL-darts | 15 | 31 | 97 |
>
> | (hours) | NS | Power | Router | C.ele | USAir | Yeast | PB | FB15k-237 | WN18RR |
> | :----: | :----: | :----: | :----: | :----: | :----: | :----: | :----: | :----: | :----: |
> | AutoGEL | 0.5 | 2.6 | 3.4 | 1.4 | 1.3 | 4.0 | 14.4 | 18.1 | 13.1 |
> | AutoGEL-darts | 1.0 | 2.7 | 3.4 | 1.5 | 1.4 | 6.0 | 14.7 | 18.3 | 13.7 |
>
> | (seconds) | IMDB-B | IMDB-M | MUTAG | PROTEINS |
> | :----: | :----: | :----: | :----: | :----: |
> | AutoGEL | 58 | 90 | 2.4 | 56 |
> | AutoGEL-darts | 122 | 138 | 3.8 | 95 |
>
> - AutoGEL alleviated the efficiency issue mentioned in the submission, i.e., using GLP is slower than some traditional models in LP tasks (e.g., bilinear models for LP on KGs).
> We can see that AutoGEL has almost the same cost as traditional NNs (or GLP) for KGs.
>
> |type| model name | NS | Power | Router | C.ele | USAir | Yeast | PB | FB15K-237 | WN18RR |
> |:----:| :----: | :----: | :----: | :----: | :----: | :----: | :----: | :----: | :----: | :----: |
> |GLP for HG| DEGNN | 0.1 | 1.0 | 1.2 | 0.2 | 0.3 | 2.0 | 4.7 | - | - |
> |bilinear for KG| DistMult | - | - | - | - | - | - | - | 2.6 | 0.4 |
> |NN for KG| ConvE| - | - | - | - | - | - | - | 26.0 | 10.2 |
> |GLP for KG| CompGCN | - | - | - | - | - | - | - | 16.1 | 7.8 |
> |our| AutoGEL (search) | 0.5 | 2.6 | 3.4 | 1.4 | 1.3 | 4.0 | 14.4 | 18.1 | 13.1 |
> |our| AutoGEL (training) | 0.3 | 0.5 | 0.7 | 0.4 | 0.4 | 1.5 | 4.8 | 13.1 | 7.3 |
>
> **RGtXj-D2:** AutoGEL does not introduce different node labeling methods for GNN link prediction.
>
> **Reply to RGtXj-D2:**
> The node labeling method (e.g., double-radius node labeling (DRNL) [2] and distance encoding (DE) [3]) is indeed an important component towards the success of structural prediction tasks (e.g., link prediction).
> Here we clarify that
> AutoGEL presets the DE [3] as the node labeling approach for the LP task (see Sec.4.1, "AutoGEL is implemented on top of DE [3]"), due to its generality and empirically good performance.
> DRNL can be regarded as a special case for DE, where the differences between them are marginal [3].
> Both DE and DRNL work well in practice.
>
> Moreover, we have thought about the idea to incorporate this design dimension into the search space and enable it to be jointly searched with other architecture components.
> But the corresponding search cost could increase with the design dimensions increase in the search space.
> Unfortunately, sacrificing search efficiency may not be able to improve the effectiveness because DE [3] is already a powerful technique.
> Out of this concern, AutoGEL presets DE as the node labeling method to better balance between effectiveness and efficiency.
>
> We realize that we should elaborate more about the setting of the node labeling method, especially since the sentence "AutoGEL is implemented on top of DE [3]" is very vague.
> We will include the above discussion in the submission.
> Thank you for pointing it out.
>
>
> **RGtXj-D3:** The author may need to demonstrate one or two findings in searched architectures. What components of a GNN are important for the link prediction task?
>
> **Reply to RGtXj-D3:**
> Thank you for the suggestion.
> We realize that our discussions in the case study (Appendix A.2) are insufficient and will include the following discussion in Appendix A.2.
>
> Generally,
> we suppose that the composition operator $\phi(h_u,h_e)$ (see Sec.3.1.1) should be one of the most critical components for the LP task on knowledge graphs (KG).
> This operator determines the way how to compose the neighborhood embedding $h_u$ and edge embedding $h_e$ to generate the message for the center node $v$.
> Actually, the composition operator $\phi$ incorporates the scoring function design in past KG embedding models, such as subtraction for geometric models and multiplication for bilinear models.
> From experiments, we observed that $\phi$ is data-dependent. $corr$ is more preferred for the FB15k\-237 dataset, and simpler $mult$ is prone to get better results for the WN18RR dataset.
> Using others $\phi$ for these data sets would lead to significantly different performance based on the empirical study.
>
>
> ## References
>
> [1] Understanding and Simplifying One-Shot Architecture Search. ICML 2018.
>
> [2] Link prediction based on graph neural networks. NeurIPS 2018.
>
> [3] Distance Encoding--Design Provably More Powerful GNNs for Structural Representation Learning. NeurIPS 2020.

---

### Official Review · Reviewer_K8wz · 2021-07-16

**Rating:** 6
**Confidence:** 4

**Summary:**

This paper proposes an automated Graph Neural Network (GNN) framework for link prediction tasks on graphs.
The experiment results suggest that the architecture search is useful.

**Limitations And Societal Impact:**

No.
Regarding the limitations of the paper, the authors mention that " In practice, GLP models run longer than Geometric and Bilinear models, which leads to the difficulty of tuning hyper-parameters". I think this is a non-trivial limitation. How difficult it is to use AutoGLP?

**Main Review:**

This paper performs a GNN architecture search for link prediction tasks.
The proposed search space is specialized for link prediction tasks.
The experiment results suggest that the architecture search is useful, however it seems that the performance gain is marginal.
In terms of novelty, the framework for the proposed search space follows paper [1], with some specialization on link prediction tasks.
Overall, the paper targets a useful problem in the graph learning community.
My main concerns are over (1) if the technical novelty is enough; (2) if the experimental results are sufficient.

Additionally, I have these specific comments for the authors to stress.

(1) The search algorithm being used is very vague (Section 3.2).
The paper claims that "AutoGEL adopts its stochastic differentiable search algorithm".
However, for discrete architectural choice, e.g., sum, mean, max for AGG, it is unclear how that is differentiable.
The authors should elaborate more on the search algorithm; otherwise, the correctness/novelty of the algorithm cannot be evaluated.

(2) In paper [1], in fact, they have done an architecture search for link prediction tasks. That part of the results is shown in the Appendix (https://arxiv.org/pdf/2011.08843.pdf). The authors should correct the descriptions in Table 1 and corresponding discussions. Additionally, the author may want to emphasize more on the specialty of the proposed search space, compared to a standard GNN search space.

(3) Regarding the limitations of the paper, the authors mention that " In practice, GLP models run longer than Geometric and Bilinear models, which leads to the difficulty of tuning hyper-parameters". I think this is a non-trivial limitation. How difficult it is to use AutoGLP?

(4) I think the authors should report one important metric, that is how many models have been searched to get each of the experiment result. The AutoML approach naturally spends much more computing resources, therefore directly comparing the numbers is not meaningful.

[1] Jiaxuan You, Zhitao Ying, and Jure Leskovec. Design space for graph neural networks. Advances in Neural Information Processing Systems, 33, 2020.

**Time Spent Reviewing:**

2

---

> ### Author Response · Authors · 2021-08-10
> **We replied to the weaknesses and detailed questions raised by the reviewer K8wz, including more discussions and more experimental reports.**
>
> We sincerely thank the reviewer’s efforts in reviewing our submission, and our detailed replies are as follows:
>
> ## Replies to Weaknesses
> **RK8wz-W1-W2:** W1: if the technical novelty is enough; W2: if the experimental results are sufficient.
>
> **Reply to RK8wz-W1-W2:**
> We suppose that the reviewer elaborated his/her concerns about W1 and W2 in detailed questions, i.e., D1 and D2 are for W1, D3 and D4 are for W2. Therefore, please refer to our following replies.
>
> ## Replies to Detailed Questions
>
> **RK8wz-D1:** The search algorithm being used is very vague. The authors should elaborate more on the search algorithm.
>
> **Reply to RK8wz-D1:**
> Thanks for pointing it out.
> We did not elaborate too much on the search algorithm due to the space limit.
> But we realize that the description of the search algorithm is insufficient and will include the following discussion in the paper.
>
> The GNN search problem is formulated as:
> $$
> max_{\theta,\omega} f(\theta,\omega;D),
> $$
> where $f(\theta,\omega;D)$ evaluates the performance of a GNN model $\theta$ with weight $\omega$ on the data set $D$.
> To solve this problem,
> we need to compute the gradients of network weight $\nabla_{\omega}f$ and GNN model $\nabla_{\theta}f$.
> Computing $\nabla_{\omega}f$ is simple.
> Unfortunately,
> the GNN model $\theta$ is discrete as reviewer K8wz pointed out, thereby $\nabla_{\theta}f$ does not exist.
> Therefore, we reformulate the problem into:
> $$
> \max_{\alpha,\omega} E_{\theta \sim p_\alpha(\theta)}[f(\theta,\omega;D)],
> $$
> where $\alpha$ is the structure parameter, $\theta \sim p_\alpha(\theta)$ represents a GNN model $\theta$ being sampled from the distribution $p_\alpha(\theta)$ that parameterized by $\alpha$, $E[\cdot]$ is the expectation.
> To compute the gradient w.r.t. $\alpha$ for the GNN model,
> we first utilize the ** reparameterization trick** $\theta = g_\alpha(U)$ presented in Eq. (12) (see Sec. 3.2 in the submission), where $U\sim p(U)$ and $p(U)$ is a uniform distribution.
> Then the gradient w.r.t. $\alpha$ is computed as:
> $$
> \nabla_{\alpha} E_{\theta \sim p_\alpha(\theta)}[f(\theta,\omega;D)]
> =
> \nabla_{\alpha} E_{U \sim p(U)}[f(g_\alpha(U),\omega;D)]
> =
> \nabla_{\alpha} \int p(U)f(g_\alpha(U),\omega;D) dU
> $$
> $$
> =
> \int p(U)\nabla_{\alpha}f(g_\alpha(U),\omega;D) dU
> =
> E_{U \sim p(U)} [\nabla_{\alpha}f(g_\alpha(U),\omega;D)]
> =
> E_{U \sim p(U)} [f'(g_\alpha(U),\omega;D)\nabla_{\alpha}g_\alpha(U)],
> $$
> where $\nabla_{\alpha}g_\alpha(U)$ can be computed because $g_\alpha(U)$ presented in Eq. (12) is differentiable.
>
> Overall,
> AutoGEL adopts
> the reparameterization trick $\theta = g_\alpha(U)$ in
> stochastic differentiable search algorithm SNAS [1], which leverages the concrete distribution from [2] to relax the discrete architecture distribution to be continuous.
>
> **Tips for Reply to RK8wz-D1**
> - $\theta$, $\alpha$, $U$ are same notations with Eq.(12) in the submission;
> - The gradients w.r.t $\omega$ can be approximated by MC sampling ($\theta^i$ represents a concrete GNN model being sampled):
> $$
> \nabla_{\omega} E_{\theta \sim p_\alpha(\theta)}[f(\theta,\omega;D)]
> = E_{\theta \sim p_\alpha(\theta)}[\nabla_{\omega}f(\theta,\omega;D)]
> \approx
> \frac{1}{N} \sum_{i} \nabla_{\omega}f(\theta^i,\omega;D)
> $$
> - The gradients w.r.t $\alpha$ after the reparameterization trick can also be approximated by MC sampling;
> - AutoGEL utilizes the bi-level formualation in practice. It uses validation data $D_{val}$ to update $\alpha$ and training data $D_{tra}$ to update $\omega$.
>
>
> **RK8wz-D2:** The paper [3] has done an architecture search for link prediction (LP) task. Besides, the author may need to emphasize more on the specialty of the proposed search space, compared to a standard GNN search space.
>
> **Reply to RK8wz-D2:**
> Thanks for pointing it out. We should correct the row "You et al. 2020" and column "Task" of Tab. 1 from "node/graph" to "node/edge/graph".
>
> Here we summarize the specialty of the proposed search space and should include the following discussion in the paper.
> AutoGEL incorporates more operation designs to further handle link information (see Sec. 3.1).
> In Sec. 3.1.1, AutoGEL introduces several operators in the intra-layer message passing design.
> First,
> the linear transformation matrix $W_\delta(u)$ is designed to distinguish edges between self-type and
> neighbor-type in the homogenous graph.
> Second,
> the linear transformation matrix
> $W_\lambda(e)$ and
> composition operator $\phi$ are designed for multi-relational graphs.
> Especially,
> the search design on
> $\phi(h_u,h_e)$ enables AutoGEL to capture semantic meaningful edges in knowledge graphs by $h_e$, and the interaction between nodes with edges by $\phi(\cdot)$.
> That is why AutoGEL can handle the LP task on both homogenous graphs and knowledge graphs, but another edge-level model You et al. 2020[3] can only cover the LP task on homogenous graphs.
>
> Besides, as discussed in Sec. 3.1.3,
> AutoGEL provides powerful pooling operators $\mathbf{h}\_{x} = R(\{\mathbf{h}\_v | v\in \mathcal{X}\})$
> for LP task on homogenous graphs, which are not included in other GLPs or AutoGLPs.
> Specifically,
> we found that DEGNN [4] presets difference-pooling for $R(\cdot)$ in the edge-level task, but it does not achieve competitive performance in the empirical study.
> But AutoGEL achieves better performance, which is demonstrated in the comparison between DEGNN with AutoGEL in Tab. 2, AutoGEL-diff with AutoGEL in Tab. 11.
>
>
> **RK8wz-D3:** The author mentions that "In practice, GNN for Link Prediction (GLP) models run longer than Geometric and Bilinear models, which leads to the difficulty of tuning hyper-parameters".
> How difficult it is to use AutoGLP?
>
> **Reply to RK8wz-D3:**
> In practice, we found that existing GLP modes generally require more computational resources.
> Thus, we try to reduce the search cost in the proposed AutoGEL.
> In the submission, we report the search cost of AutoGEL in the LP task in Tab. 9 (see Appendix A.3).
> Here we further extend this table to show the difficulty in using AutoGEL for LP tasks.
> The table below reports
> the running time (hours) of AutoGEL and several other representative baselines for the LP task on the homogenous graph (HG) and knowledge graph (KG).
>
> |type|model name|NS|Power|Router|C.ele|USAir|Yeast|PB|FB15K-237|WN18RR|
> |:----:|:----:|:----:|:----:|:----:|:----:|:----:|:----:|:----:|:----:|:----:|
> |GLP for HG| DEGNN[4] | 0.1 | 1.0 | 1.2 | 0.2 | 0.3 | 2.0 | 4.7 | - | - |
> |bilinear for KG| DistMult[5] |-|-|-|-|-|-|-|2.6|0.4|
> |NN for KG| ConvE[6]|-|-|-|-|-|-|-|26.0|10.2|
> |GLP for KG| CompGCN[7] |-|-|-|-|-|-|-|16.1|7.8|
> |our| AutoGEL (search) |0.5|2.6|3.4|1.4|1.3|4.0|14.4|18.1|13.1|
> |our| AutoGEL (training) |0.3|0.5|0.7|0.4|0.4|1.5|4.8|13.1|7.3|
>
> From the above table, we can observe that:
> - On the LP task on HGs (NA, Power, Router, C.ele, USAir, Yeast, and PB), AutoGEL runs quite fast, which substantially eases the difficulty of using AutoGEL. Besides, AutoGEL achieves more significant performance boost in this scenario (see Tab.2).
> - On the LP task on KGs (FB15K-237, WN18RR), DistMult[5] is a representative of bilinear models that run much faster among all KGE models. Although AutoGEL is slower than DistMult, its computational cost is very close with classic neural networks (NNs) for KG ConvE[6] and GLP model CompGCN[7]. Then recalling Tab. 3 in submission, AutoGEL well balances between search cost and effectiveness.
>
>
> **RK8wz-D4:** How many models have been searched to get each of the experiment results. The AutoML approach naturally spends much more computing resources, therefore directly comparing the numbers is not meaningful.
>
> **Reply to RK8wz-D4:**
> As reviewer K8wz pointed out,
> AutoML methods may run many times, then select a searched architecture with the highest performance, which is unfair to other non-searching methods.
> Out of this concern,
> all effectiveness results (Tab. 2, Tab. 3, and Tab. 4) are reported under the average of 4 runs.
> Note that Tab. 3 and Tab. 4 do not contain variance due to space limits.
> To illustrate the "worst" case of AutoGEL, here we report results after multiple runs of AutoGEL on several tasks:
>
> In Tab. 2 of submission, AutoGEL reports $99.89\pm 0.06$ for the LP task on the NS dataset. And there are results of 50 runs:
>
> |result interval|99.80-99.85|99.85-99.90|99.90-99.95|99.95-100.00|
> |:----:|:----:|:----:|:----:|:----:|
> |frequency|3|20|17|10|
>
> In Tab. 3 of submission, AutoGEL reports $.357$ (MRR) for the LP task on the FB15k\-237 dataset. And there are results of 10 runs (because of the short rebuttal period):
>
> |result interval|.354-.355|.355-.356|.356-.357|.357-.358|.358-.359|
> |:----:|:----:|:----:|:----:|:----:|:----:|
> |frequency|3|2|4|0|1|
>
> In Tab. 4 of submission, AutoGEL reports $89.89$ for the node classification task on the Cora dataset. And there are results of 50 runs:
>
> |result interval|89.0-89.5|89.5-90.0|90.0-90.5|90.5-91.0|91.0-91.5|91.5-92.0|
> |:----:|:----:|:----:|:----:|:----:|:----:|:----:|
> | frequency | 2 | 19 | 16 | 10 | 2 | 1 |
>
> In Tab. 4 of submission, AutoGEL reports $82.68$ for the graph classification task on the PROTEINS dataset. And there are results of 50 runs:
>
> | result interval | 80.0-81.0 | 81.0-82.0 | 82.0-83.0 | 83.0-84.0 |
> |:----:| :----: | :----: | :----: | :----: |
> | frequency | 25 | 14 | 9 | 2 |
>
> Thanks for reminding us.
> We should include the above discussion in all graph tasks and data sets in the
> appendix.
>
>
>
> ## References
>
> [1] SNAS: stochastic neural architecture search. ICLR 2019.
>
> [2] The concrete distribution: A continuous relaxation of discrete random variables. ICLR 2017.
>
> [3] Design space for graph neural networks. NeurIPS 2020.
>
> [4] Distance Encoding--Design Provably More Powerful GNNs for Structural Representation Learning. NeurIPS 2020.
>
> [5] Embedding entities and relations for learning and inference in knowledge bases. ICLR 2015.
>
> [6] Convolutional 2d knowledge graph embeddings. AAAI 2018.
>
> [7] Composition-based multi-relational graph convolutional networks. ICLR 2020.

---

> > ### Comment · Reviewer_K8wz · 2021-08-27
> > **Reply to the authors**
> >
> > I appreciate the authors for adding the explanations for the search algorithm and providing many new experimental results. I suggest that these descriptions and results should be included in the revised version. I would like to raise my score to 6.

---

> > > ### Author Response · Authors · 2021-08-28
> > > **Reply to the Reviewer RK8wz**
> > >
> > > We sincerely appreciate your comments. And we will definitely incorporate new discussion and experimental results in the final version of the paper.

---

### Official Review · Reviewer_Gqb4 · 2021-07-17

**Rating:** 5
**Confidence:** 3

**Summary:**

The paper introduces an automated graph neural network model with explicit link information. The model is tested on link prediction task (both homogeneous and heterogenous graphs) and node classification task. It shows improvement compared with the baseline models selected.

**Limitations And Societal Impact:**

I don't see any potential negative societal impact of their work.

**Main Review:**

The paper mostly focuses on explaining how it incorporates the link information (edges) and it can generate edge embeddings so it would be suitable for link prediction in multi-relational graphs, such as knowledge graphs. The proposed method is primarily based on existing approaches with a few incremental tweaks and combining different components from different approaches (e.g., combining intra-layer message passing and inter-layer message passing). However, in the experiments, it seems that the model has yet being compared with the most up-to-date models in the link prediction task with FB15k-237 dataset (https://paperswithcode.com/sota/link-prediction-on-fb15k-237). So the overall contribution is unclear. Although link prediction is a hard task considering that there are now a lot of high baseline models to compare, the selling point of this paper seems to be using link information to improve upon link prediction task. The paper does not seem to convince the readers on this aspect.

**Time Spent Reviewing:**

2

---

> ### Author Response · Authors · 2021-08-10
> **We answered why some models of the LP task on FB15k-237 are not included in this paper.**
>
> We sincerely thank the reviewer’s efforts in reviewing our submission, and our detailed replies are as follows:
>
>
> ## Replies to Weakness
>
> **RGqb4-W1:**
> In the link prediction task on knowledge graph (KG) with FB15K-237 dataset, the authors do not include the most state-of-the-art baselines in https://paperswithcode.com/sota/link-prediction-on-fb15k-237.
>
> **Reply to RGqb4-W1:**
> It has been discussed in two published works https://arxiv.org/pdf/1911.03903.pdf [1] and https://dl.acm.org/doi/abs/10.1145/3424672 [2] that the evaluation of these high baseline models (e.g., ConvKB[3], CapsE[4], and KBGAT[5])
> are unfair and implausible because of the **Tie Policy** they adopted.
>
> In the link prediction task, KG embedding (KGE) models target to predict the missing entity in given triplet $<h,r,?>$ or $<?,r,t>$.
> Generally, all candidate entities will be sorted in descending order based on scores computed by KGE models.
> For example, five candidate entities $\\{e_1, e_2, e_3, e_4, e_5\\}$ have corresponding scores as $[2,3,1,4,5]$. Then their rankings will be: $\\{e_1:4, e_2:3, e_3:5, e_4:2, e_5:1\\}$.
> KGE models want to give a larger score of ground truth entity, i.e., let the ranking value of the target entity be smaller. If $e_3$ is the ground truth in this example, its ranking 5 implies a failed embedding result.
>
> The most common evaluation matrics for the LP task on KG are MRR and HIT@N, which are based on the rankings of the target entity.
> - Mean Reciprocal Ranking (MRR). It is the average of the inverse of the obtained ranks: $(\sum_{i=1}^{|S|}1/rank_i)/|S|$, where $S$ and $rank_i$ denote test triples and ranking results;
> - HIT@N. It is the ratio of predictions for which the rank is equal or lesser than a threshold N: $(\sum_{i=1}^{|S|}\mathbb{1}(rank_i\leq{N}))/|S|$, where $\mathbb{1}$ denotes indicator function, and $N\in\\{1,3,10\\}$.
> In MRR and HIT@N, it can be seen that the smaller the ranking value of the target entity, the higher the evaluation performance.
>
> **Tie Policy** matters when the scores of candidate entities are the same. For example, how to assign their ranking values when five candidates have the same scores $[0,0,0,0,0]$?
> Currently, there are several policies adopted in KGE models:
> - **min (top)**: the target is given the best rank among the entities in a tie. This is the least strict policy, and it may result in artificially boosting performances.
> In this example, the ranking value of ground truth will be 1 when the KGE model does not know anything, which leads to MRR = 1 and HIT@N = 1.
> - **max (bottom)**: the target is given the worst rank among the entities in a tie. This is the most strict policy. In this example, the ranking value of ground truth will be 5, thereby MRR = 0.2, HIT@1 = 0, HIT@3 = 0.
> - **average**: the target is given the average rank among the entities in a tie.
> In this example, the ranking value of ground truth will be 3, which leads to MRR = 0.33, HIT@1 = 1, HIT@3=1.
> - **random**: the target is given the random rank among the entities in a tie.
>
> Generally, a KGE should be able to differentiate positive triples from their negative samples and thus be robust to different tie policies.
> However, it has been observed in [1][2] that these works with high reported performance in https://paperswithcode.com/sota/link-prediction-on-fb15k-237 (e.g., ConvKB[3], CapsE[4], and KBGAT[5]) have huge performance discrepancies when using different policies (please check Tab. 2 in [1] and Tab. 4 in [2]).
> But other baselines (e.g., ConvE and RotatE adopted in our submission) could keep consistent results.
> That is mainly because these models with high report performance (ConvKB[3], CapsE[4], and KBGAT[5])
> are prone to give the same score to different entities, such as using the ReLU function to output final scores of candidate entities (see Sec.3.2 in [1] and Sec.5.5 in [2]).
> Note that the top-1 baseline in https://paperswithcode.com/sota/link-prediction-on-fb15k-237 is GAATs[6], which has not been discussed in literature due to no publicly available code.
> But GAATs[6] utilizes CapsE[4] as its decoder, which computes the scores of candidate entities.
> In other words, GAATs also tends to give the same score to different entities like CapsE, i.e., it probably has the same tie policy issue.
>
> In this paper, we report our results based on the average policy, which is recommended in [1][2].
> And our model is robust to other policies.
> Thus, the models discussed above are excluded from our baselines for fair comparison.
>
> ### Tips
> - KBGAT[5] is named as KBAT in [1].
>
>
> ## References
>
> [1] A re-evaluation of knowledge graph completion methods[J]. ACL 2020. https://arxiv.org/pdf/1911.03903.pdf
>
> [2] Knowledge graph embedding for link prediction: A comparative analysis[J]. TKDD 2021. https://dl.acm.org/doi/abs/10.1145/3424672
>
> [3] A Novel Embedding Model for Knowledge Base Completion Based on Convolutional Neural Network. NAACL 2018.
>
> [4] A capsule network-based embedding model for knowledge graph completion and search personalization. ACL 2019.
>
> [5] Learning Attention-based Embeddings for Relation Prediction in Knowledge Graphs. ACL 2019.
>
> [6] Knowledge Graph Embedding via Graph Attenuated Attention Networks.

---

### Official Review · Reviewer_q5RS · 2021-07-17

**Rating:** 6
**Confidence:** 3

**Summary:**

The authors propose an AutoGNN model that introduces more design dimensions in both inter- and intro-layer. It is a decent attempt for authors to take into consideration the edge embedding in MPNN and adopt a stochastic differentiable search algorithm, making the model be able to better handle the link prediction problem and get more efficiently optimized.

**Ethical Concerns:**

There are no ethical concerns.

**Limitations And Societal Impact:**

There does not appear to be any negative societal impact of this work. In fact, it helps alleviates some of the impracticalness of some prior autoGNN work regarding the computational running time needed for slight performance improvements (when thinking from an environmental standpoint).

**Main Review:**

Strengths:

S1) The paper is well-written and organized, and the proposed model (AutoGEL) is technically sound. In AutoGEL, more design spaces are introduced to be learned both with intra- and inter-layer, which would intuitively make the trained model perform better than manually designed ones.

S2) In AutoGEL authors explicitly learns the edge embedding in the message passing framework to model the complex link information, making AutoGNN-like model be generalized to the LP task.

S3) Adopting differentiable search algorithm to optimize the GNN architecture parameters makes the model be able to be trained more efficiently and stably.

S4) Extensive experiments are well organized to show the effectiveness and competitiveness of AutoGEL.


Weaknesses:

W1) As authors mentioned in Section 4.2, the more the design dimensions, the longer the model will run, which indeed would be the case. Although the running times are still rather low, explicitly showing the scalable would be worth including.

W2) There are still critical structure components needing to be manually designed such as the number of layers of the model, which would have significant impact on the performance of model even after GNN operators being decided.

W3) The problem is not as novel, but Table 1 did help frame and clarify the distinction of this work compared to other recent related autoGNN methods.


Other Comments/Questions/Typos:

1) In the paper, the authors attempt to add edge embedding in MPNN to handle the LP task and also improve the performance of the model on other tasks. However, in the experiments section, it seems in the specific models that are used in node classification and graph classification are missing the edge embedding component (as demonstrated in Figures 4 and 5 in the appendix).

2) In the ablation study, it seems better to add some experiments for testing the effectiveness of the edge embedding. (i.e., using model without the edge embedding component on LP tasks).

3) "The empirical comparison on the node classification and graph classification task is shown in Table 7." is mentioned, but there is no Table 7 and it also does not correspond to Table 7 in the supplemental file.

4) For selecting an operation in an operator, would it be necessary for the operators to converge to a one-hot vector? What if not and just take the real-valued operator vector as a mixed strategy?

5) How does the edge embedding matrix (i.e., h_e^x) update?

6) I was thinking more design dimensions can be similarly regarded as more parameters (but in terms of structure not weight), thus it would somewhat make it easier for overfitting.

Please note that I have gone through the author response and other reviews/comments, but respectfully keep my score of 6 (i.e., unchanged).


**Time Spent Reviewing:**

4

---

> ### Author Response · Authors · 2021-08-10
> **We replied to the weaknesses and detailed questions raised by the reviewer q5RS, including more discussions and more experimental reports.**
>
> We sincerely thank the reviewer’s efforts in reviewing our submission, and our detailed replies are as follows:
>
>
> ## Replies to Weaknesses
> **Rq5RS-W1:** The scalability of design dimensions needs to be further discussed.
>
> **Reply to Rq5RS-W1:**
> As discussed, such scalability issue generally exists in searching models.
> The more the design dimensions, the longer the search time.
> That is because the larger search space has more architectures to be evaluated during the search.
> One principle in neural architecture search can partially alleviate this scalability issue, i.e., a good space should be expressive enough to enable some powerful models to be searched but it should not be so large that the search cost is too high.
>
> In this paper, we follow the principle to carefully design AutoGEL's space.
> This space generally contains powerful components in existing GNN designs (e.g., composition operator) and avoids some components that cannot achieve good performance or require too many computational resources (may refer to Rq5RS-W2 for more discussions).
> And we demonstrate the effectiveness and efficiency of AutoGEL in the empirical study.
>
> At the same time, we also realize that the discussion of scalability in this paper is insufficient.
> We need to show some experiments that adding some design dimensions/operators may increase the search cost but cannot achieve better performance.
>
>
> **Rq5RS-W2:** As one of the crucial components in GNNs, the number of layers needs to be manually designed.
>
> **Reply to Rq5RS-W2:**
> In this paper, we did not take the number of GNN layers as a search dimension because of the trade-off between efficiency and effectiveness.
>
> - Intuitively, including one design dimension could increase the size of search space, which leads to more computational cost in searching. However, it has been studied that most of the successful GNNs are still shallow due to the over-smoothing issue [1] brought by deeper networks. In other words, sacrificing search efficiency may not be able to improve the effectiveness. Therefore, we empirically set the number of layers as a small fixed value, which is also a common practice for most of the existing AutoGNNs.
> - Moreover, AutoGraph[2] leverages an evolutionary algorithm to enable searching the number of layers. However, AutoGraph takes almost $10^3$ magnitude longer time than AutoGEL (may refer to AutoGraph's Tab.4 and AutoGEL's Tab.5 for comparison) while still limited to marginal performance gains compared with baselines.
> - Lastly, AutoGEL can be easily extended to search for the optimal depth. We would like to have a try on it if the community has explored some techniques to alleviate the over-smoothing issue.
>
>
> **Rq5RS-W3:** The problem is not as novel, but Tab. 1 helps frame and clarifies the distinction between related works.
>
> **Reply to Rq5RS-W3:**
> Yes, this paper is built on literature, thus we utilize Tab. 1 to show the difference between our designs and related works.
>
>
>
> ## Replies to Detailed Questions
>
> **Rq5RS-D1:** It seems the edge embedding is missing in node/graph classification tasks as shown in Fig. 4 and 5.
>
> **Reply to Rq5RS-D1:**
> First, we clarify that the overall idea of AutoGEL is to leverage the link information.
>
> This question is related to the text description below Eq. (7)-(9) in Sec. 3.1.1.
> AutoGEL adopts different ways to encode link information for two graph types.
> Eq. (7) uses $W_{\delta}$ to encode link information, while Eq. (8) uses $W_{\lambda}$ and the edge embedding $h_{e}$.
>
> The benchmark datasets adopted for node/graph classification tasks usually do not contain semantic meaningful edges (unlike knowledge graphs).
> Those edges only represent whether connections between $u$ and $v$ (or $v$ and $v$ by adding self-loop) exist or not.
> Therefore, it is unnecessary to introduce edge embedding in node/graph classification tasks, i.e.,
> $W_{\delta}$ in Eq. (7) is enough to capture the link information in these two tasks.
> By introducing $W_{self}$ and $W_{neigh}$ ($\delta(u) \in \\{self, neigh\\}$) for the node itself and neighbors separately, AutoGEL is able to differentiates different impact between information from self-loop edges and regular edges.
>
> **Rq5RS-D2:** It is better to add the ablation study on the edge embedding.
>
> **Reply to Rq5RS-D2:** Thanks for the suggestion!
>
> To show the effectiveness of edge embedding $h_e$, we set another variant, i.e., AutoGEL-$\setminus h_e$, by removing $h_e$ from AutoGEL's MPNN and simply replace $\phi(h\_u^k,h\_e^k)$ with $h_u^k$ in Eq. (8).
> Experiment results on the WN18RR dataset are shown below.
> Significant performance degradation is observed for the AutoGEL-$\setminus h_e$, indicating that $h_e$ is indeed a critical design.
>
> ||MRR|Hits@10|Hits@3|Hits@1|
> |:----:|:----:|:----:|:----:|:----:|
> |AutoGEL|.479 |.549 |.492 |.444 |
> |AutoGEL-$\setminus h_e$|.454 |.540|.483 |.402 |
>
>
>
> **Rq5RS-D3:** There is a typo in "The empirical comparison ... in Tab. 7"
>
> **Reply to Rq5RS-D3:** Thank you for pointing out the typo. It should be Tab. 4 here.
>
>
> **Rq5RS-D4:** Would it be necessary for the operators to converge to a one-hot vector? What if not and just take the real-valued operator vector as a mixed strategy?
>
> **Reply to Rq5RS-D4:** Here we explain the reason why AutoGEL adopts the stochastic differentiable algorithm SNAS[3] (i.e., the one-hot vector way) instead of the deterministic differentiable algorithm (i.e., mixed strategy).
>
> The mixed strategy (e.g., representative work DARTS[4]) relaxes the discrete architecture distribution to be continuous, and thus makes the searching process deterministic differentiable.
> However, several drawbacks brought by such mixed strategy have been observed and discussed in the community of neural architecture search:
> - The one-hot strategy is better than the mixed strategy from the perspective of effectiveness. The mixed strategy usually leads to the inconsistent performance issue, i.e., the performance of the derived child network at the end of the searching stage shows significant **degradation** compared with the performance of the parent network before architecture derivation.
> But the one-hot strategy could keep performance consistency, thanks to the discrete architecture selection during forward propagation.
> [3] provides theoretical proof that the stochastic differentiable search algorithm of SNAS is unbiased once converged, while mixed strategy would bring much more bias towards the optimization objective.
> - The one-hot strategy is better than the mixed strategy from the perspective of efficiency.
> During the search, the mixed strategy must maintain all operators in the whole supernet, which requires more computational resources than the one-hot vector. This part has been discussed in [5].
>
> Moreover, we have a similar observation in our GNN search scenario. Tab. 11-13 in the Appendix empirically shows the consistent inferior performance of the AutoGEL-darts variant (by replacing AutoGEL's search algorithm with mixed strategy) cross node/edge/graph level tasks.
> And here we also show search cost of AutoGEL-darts is higher than AutoGEL:
>
>
> |(seconds)|Cora|CiteSeer|PubMed|
> |:----:|:----:|:----:|:----:|
> |AutoGEL|12|16|19|
> |AutoGEL-darts|15|31|97|
>
> |(hours)|NS|Power|Router|C.ele|USAir|Yeast|PB|FB15k-237|WN18RR|
> |:----:|:----:|:----:|:----:|:----:|:----:|:----:|:----:|:----:|:----:|
> |AutoGEL|0.5|2.6|3.4|1.4|1.3|4.0|14.4|18.1|13.1|
> |AutoGEL-darts|1.0|2.7|3.4|1.5|1.4|6.0|14.7|18.3|13.7|
>
> | (seconds) | IMDB-B | IMDB-M | MUTAG | PROTEINS |
> | :----: | :----: | :----: | :----: | :----: |
> | AutoGEL | 58 | 90 | 2.4 | 56 |
> | AutoGEL-darts | 122 | 138 | 3.8 | 95 |
>
>
> Thanks for reminding us. We will add the above discussion into Sec. 3.2 to strengthen the motivation of adopted search algorithm and table content in Tab. 9-10.
>
>
> **Rq5RS-D5:** The edge embedding updation is missing.
>
> **Reply to Rq5RS-D5:** Thank you for pointing out. We should include this part in the main text.
>
> The edge embedding matrix $h^k_e$ in Eq. (8) is updated through linear transformation $W^k_{rel}$:
> $$h^{k+1}\_e=W^k_{rel}h^k_e$$
>
>
> **Rq5RS-D6:** Can more design dimensions be regarded as some structure parameters? Is there any over-fitting problem?
>
> **Reply to Rq5RS-D6:** Due to the space limit, we did not include the overall problem formulation in the submission.
> Here we introduce it as follows:
>
> $$
> \alpha^* = \max_{\alpha} E_{\theta \sim p_\alpha(\theta)}[f(\theta,\omega^*;D_{val})], \\
> s.t. \omega^* = \max_{\omega} f(\theta,\omega;D_{tra}),
> $$
> where $\alpha$ is the structure parameter, $\omega$ is the network weight, $\theta \sim p_\alpha(\theta)$ represents a GNN model $\theta$ being sampled from the distribution $p_\alpha(\theta)$ that parameterized by $\alpha$, $E[\cdot]$ is the expectation, $f(\theta,\omega;D)$ evaluates the performance of model $\theta$ with weight $\omega$ on the data set $D$.
>
> - For the first question, the design dimension indeed can be regarded as structure parameters. We follow the classic bi-level formulation in neural architecture search, where the upper level is to optimize the parameters of structures $\alpha$ and the lower level is to optimize network weight $\omega$.
>
> - For the second question, we utilize training data $D_{tra}$ to optimize network weight $\omega$ and validation data $D_{val}$ to optimize structure parameter $\alpha$, which guarantees the generalization ability of AutoGEL and avoids the overfitting issue.
>
> We realize that we should include the above formulation and discussion in the appendix. Thanks for reminding us.
>
>
>
> ## References
>
> [1] Deeper insights into graph convolutional networks for semi-supervised learning. AAAI 2018.
>
> [2] Autograph: Automated graph neural network. ICONIP 2020.
>
> [3] SNAS: stochastic neural architecture search. ICLR 2019.
>
> [4] Darts: Differentiable architecture search. ICLR 2019.
>
> [5] Efficient neural architecture search via proximal iterations. AAAI 2020.

---

> > ### Comment · Reviewer_q5RS · 2021-09-02
> > **Response to authors**
> >
> > First, thank you for your detailed response. We especially appreciate you looking into the ablation study on the edge embeddings, clarifying the update of the edge embedding, and the comparison with your variant AutoGEL-darts. Note that I am still positive of this paper, but respectfully keep my score "6: Marginally above the acceptance threshold" unchanged.

---

> > > ### Author Response · Authors · 2021-09-03
> > > **Response to Reviewer q5RS**
> > >
> > > We sincerely thank your efforts in reviewing our submission :)

---

### Decision · Program_Chairs · 2021-09-27

**Decision:**

Accept (Poster)

**Comment:**

This paper proposes a new autoGNN model that takes link information into considerations. The model has shown better performance compared to baselines. The reviewers had concerns on novelty and baselines, and suggested new experiments. The authors have done a good job in addressing reviewers’ concerns, which is reflected in the increase of some rating. The concerns regarding to KG link prediction tasks has been well addressed in the rebuttal, although the reviewer didn’t raise their rating. Overall, the paper extends an existing algorithm, by considering link prediction task and modeling edge information. Although the idea is incremental, the execution of this paper is solid. I thus recommend accept.